# Optimizing Time Series Forecasting Architectures: A Hierarchical Neural Architecture Search Approach

**Difan Deng**  *d.deng@ai.uni-hannover.de*
*Institute of Artificial Intelligence*
*Leibniz University Hannover*

**Marius Lindauer**  *m.lindauer@ai.uni-hannover.de*
*Institute of Artificial Intelligence*
*Leibniz University Hannover*
*L3S Research Center*

**Reviewed on OpenReview:** *https://openreview.net/forum?id=Ym2wqojm4e*

## Abstract

The rapid development of time series forecasting research has brought many deep learning-based modules to this field. However, despite the increasing number of new forecasting architectures, it is still unclear if we have leveraged the full potential of these existing modules within a properly designed architecture. In this work, we propose a novel hierarchical neural architecture search space for time series forecasting tasks. With the design of a hierarchical search space, we incorporate many architecture types designed for forecasting tasks and allow for the efficient combination of different forecasting architecture modules. Results on long-term time series forecasting tasks show that our approach can search for lightweight, high-performing forecasting architectures across different forecasting tasks.

## 1 Introduction

Time series forecasting techniques are widely applied in different fields, e.g., energy consumption (Trindade, 2015), business (Makridakis et al., 2022), or traffic planning (Lana et al., 2018). However, unlike computer vision (CV) and natural language processing (NLP) tasks that are dominated by the CNN (He et al., 2016; Liu et al., 2018; Zoph et al., 2018) and Transformer (Brown et al., 2020; Devlin et al., 2019; Liu et al., 2021; Vaswani et al., 2017) families, there is no clearly dominating architecture in time series forecasting tasks. Although there are lots of transformer-based approaches applied to forecasting models (Ansari et al., 2024; Das et al., 2023; Wu et al., 2021; Liu et al., 2022; Zhou et al., 2021), many of them might even be outperformed by a simple linear baseline (Zeng et al., 2023).

The success of transformer models in CV and NLP tasks is based on their ability to capture long-term dependencies with the help of tokens with fruitful semantic information, i.e., each word embedding already contains lots of information, while an image patch with many pixels can already tell us a lot of information. All this information relaxes the requirement for the models to grab the temporal information within the input series. However, this information is usually crucial in time series forecasting tasks, given that a single value at each time step only contains a limited amount of information. The strong ability of the transformer family to capture long-term dependencies might not compensate for its poor ability to build temporal connections within the input series. This finding is evident in Zeng et al. (2023), where a simple linear layer could outperform many state-of-the-art transformer models.

Some recent works show the efficiency of transformers (Liu et al., 2024b; Nie et al., 2023) on long-time forecasting tasks. However, their approach could still not overcome the temporal dependencies issues for transformers, e.g., PatchTST (Nie et al., 2023) augments the information within each token by constructing

a patch with the data from multiple time steps. While iTransformer (Liu et al., 2024b) simply encodes the entire input sequence into a token and tries to construct the connections among different variables.

On the other hand, many architectures were constructed for mining the local dependencies, such as CNNs (Bai et al., 2018; Luo and Wang, 2024) and RNNs (Hewamalage et al., 2021; Hochreiter et al., 2001). These architectures might not work well on long-term dependencies due to the limited receptive field (for CNN), latent bottlenecks (Didolkar et al., 2022), or vanishing gradients (for RNN families). Recent work, such as ModernTCN (Luo and Wang, 2024), showed that an increased convolutional kernel size and improved microarchitecture could lead to a more accurate model. However, this approach results in huge memory and computation consumption. Here we provide another perspective: one could combine these architectures with other operations, such as transformer layers, to develop a new architecture that combines the best of two worlds (Didolkar et al., 2022; Lai et al., 2018; Lim et al., 2021).

Nevertheless, it is still unclear (i) which type of architecture we would like to construct, given the variability of different forecasting models (Deng et al., 2022; Oreshkin et al., 2020; Salinas et al., 2020; Zeng et al., 2023) and (ii) how to connect different operations to form a new architecture. Designing a new architecture from scratch for each task might take a lot of human expert effort and tedious trial-and-error. Neural architecture search (NAS) is a technique that automatically searches for the optimal architecture given a new task.

Previous NAS frameworks mainly focused on single network backbone types such as CNN or Transformer networks (Chen et al., 2021b; Liu et al., 2018; Zoph et al., 2018). It is still unclear how to optimize the forecasting architectures due to their internal complexity. For instance, an encoder-decoder architecture (Wu et al., 2021; Zhou et al., 2021) might work well on some tasks, while the other tasks might prefer encoder-only architectures (Liu et al., 2024b) or even MLP-only architectures (Oreshkin et al., 2020; Zeng et al., 2023). This provides another challenge for designing a search space for time series forecasting tasks. In this work, we will address this challenge by designing a unified search space for time series forecasting tasks.

Our contributions are summarized as follows:

1. We design a hierarchical search space that contains most forecasting architecture design decisions and allows any sort of architecture layers to be combined to form new architectures.

2. We show that by applying DARTS-PT (Wang et al., 2021), a differentiable neural architecture search approach, to our search space. The resulting architectures, dubbed DARTS-TS, are comparable to the state-of-the-art models with much less computational resource requirements.

3. We provide an analysis of our search space, showing that our search space has different properties compared to the existing CNN-based NAS search space, and provides further challenges for the NAS research.

## 2 Related Work

Although many different deep learning architectures are proposed to solve time series tasks, there is little work that applies neural architecture search to search for a new architecture within the existing frameworks. In this section, we will provide a brief overview of deep learning-based forecasting frameworks and neural architecture search techniques.

### 2.1 Deep Learning-based Time Series Forecasting

Time series forecasting aims to predict the future values of target variables given their historical data. Because of its importance, much work has been investigated for a more accurate forecasting model in this research field. Previous work mainly focused on traditional statistical local approaches that train an individual model for each series (Athanasopoulos, 2021; Box et al., 2015). However, these approaches might not fit well in the era of big data, where a dataset could contain thousands of series. On the other side, the machine learning-based model trains a single global model across all the series and uses this model to predict all the series in the dataset (Godahewa et al., 2021; Makridakis et al., 2020; 2022). More recently, deep learning-based forecasting

models (Bai et al., 2018; Hewamalage et al., 2021; Lai et al., 2018; Salinas et al., 2020; Shi et al., 2015; Wen et al., 2017), or even the zero-shot foundation models (Ansari et al., 2024; Das et al., 2023), have gradually become mainstream in this research field.

Time series forecasting models need to work with sequential inputs (Alexandrov et al., 2020; Beitner, 2020). Overall, these networks can be categorized into two families: *Seq Net* and *Flat Net* (Deng et al., 2022). Given a batch of sequences with shape $[B, L, N]$, where $B$ is the batch size, $L$ is the sequence length, and $N$ is the number of time series variables. A *Seq Net*, such as RNNs (Cho et al., 2014; Hochreiter and Schmidhuber, 1997), TCNs (Bai et al., 2018; Oord et al., 2016), and Transformers (Li et al., 2019; Liu et al., 2022; Vaswani et al., 2017; Wu et al., 2021; Zhou et al., 2021), computes the correlations across different time steps without breaking the structure of the input sequence. On the contrary, a *Flat Net*, such as MLP (Zeng et al., 2023) and N-BEATS (Oreshkin et al., 2020), decomposes the variables into independent single-variable series: $[B \times N, L]$.[1]. After that, this variable is passed to another network. This strategy was previously introduced such that the time series can be handled by the machine learning framework designed for tabular datasets, such as MLP layers (Zeng et al., 2023) and LightGBM (Ke et al., 2017; Makridakis et al., 2022). Some recent works, such as PatchTST (Nie et al., 2023), release the correlation between different variables within a multi-variable series and consider it as a collection of independent series, and make predictions for each series independently, which also belongs to this type of architecture. However, these approaches might be too expensive for datasets with many variables since a forward pass is required for each series. Hence, in this work, we mainly focus on the MLP-based *Flat Net* to only search for lightweight architectures.

Diving deeper into the *Seq Net* architecture families, we find another two main branches: encoder-decoder architectures and encoder-only architectures[2]. Encoder-decoder architectures (Sutskever et al., 2014; Vaswani et al., 2017) maintain two individual networks that embed the information from the past and future correspondingly. These architectures have shown great success in time series forecasting tasks (Wu et al., 2021; Zhou et al., 2021). On the other hand, encoder-only architectures only apply a *Seq* encoder that maps the past information into a latent feature map and utilizes another linear layer to provide the prediction with the latent feature map (Liu et al., 2024b; Nie et al., 2023; Salinas et al., 2020).

Many *Seq Net* models, especially the transformer family (Lim et al., 2021; Wu et al., 2021; Zhou et al., 2021), are designed for solving series input. However, a study by Zeng et al. (2023) showed that these transformers might even be outperformed by a linear model. This inspires us to seek other uncovered modules that might perform well within a properly designed architecture, esp. architectures with more than one type of operation.

Many forecasting architectures are homogenous and only contain one type of operation layer (Bai et al., 2018; Li et al., 2019; Liu et al., 2022; Luo and Wang, 2024; Oord et al., 2016) and stack this layer repeatedly to construct a new architecture. However, some recent work has also shown the efficiency of combining architectures from different model families. LSTNet (Lai et al., 2018) stacks an RNN on top of a CNN layer. ConvLSTM (Shi et al., 2015) constructs a convolutionary operation within an LSTM cell. Temporal fusion transformer (Lim et al., 2021) stacks an explainable multi-head attention layer on top of an LSTM encoder-decoder model. All these works suggest the efficiency of combining operations from different architecture families. However, it is often tedious to find these combinations manually. Neural architecture search (NAS) is a technique that automatically searches for the optimal architecture for a given task. In the following section, we will briefly overview the NAS framework.

## 2.2 Neural Architecture Search

Previous NAS research mainly considered network training as a black-box process, training every configuration from scratch until convergence (Deng et al., 2022; Jin et al., 2019; White et al., 2021; Zimmer et al., 2021; Zoph and Le, 2017). However, training a network is very expensive and requires lots of resources. To overcome this problem, the One-Shot NAS (Pham et al., 2018; Zoph et al., 2018) approach defines a supernetwork where all the child architectures' weights are inherited from the supernet. DARTS (Liu et al., 2019a) further

---

[1]However, in practice, we would still like to preserve the correlations between multi-variant series with, for instance, batch normalization (Ioffe and Szegedy, 2015). In this case, the input series becomes $[B, N, L]$

[2]Sometimes we might also have decoder-only architectures; however, for the sake of simplicity, we consider both as part of encoder-only architectures.

relaxes the discrete operation search space to continuous parameters and optimizes these values with model weights jointly with gradient descent. Finally, the optimal operations and paths are selected based on the architecture parameter values.

DARTS may be unstable during the search process: the architectures suggested by the DARTS might be dominated by skip connections (Chu et al., 2020; Jiang et al., 2023; Wang et al., 2021; Zela et al., 2020). Robust DARTS (Zela et al., 2020) showed that this instability is caused by the high validation loss curvature in the search space, while operation-level early stopping DARTS (Jiang et al., 2023) found that the instability arises when the network weights are overfitted to the training sets. DARTS-PT (Wang et al., 2021) provides a perturbation-based approach to measure the importance of each operation and use it to replace the architecture parameter-based approach introduced in Liu et al. (2019a).

ONE-NAS (Lyu et al., 2023) is an online architecture search framework that applies evolutionary algorithms to search for the optimal RNN networks on online forecasting tasks. On the other hand, SNAS4MTF (Chen et al., 2021a) proposes to use architecture search to form an end-to-end forecasting architecture framework. Furthermore, Auto-PyTorch TS (Deng et al., 2022) provides a uniform search space that includes many forecasting modules. The optimizers can freely assemble these modules to form new architectures. However, these works still focus on homogeneous architecture designs, where the type of *Seq* decoders is restricted by the decision of *Seq* Encoders. For instance, an RNN encoder only allows RNN decoders (*Seq* decoder) or MLP decoders (*Flat* decoder). As the current neural architecture tends to focus more on the one-shot weight-sharing approaches (Jiang et al., 2023; Liu et al., 2019a; Wang et al., 2021; Zoph et al., 2018), in this work, we will show how to design a general one-shot model for time series forecasting that fits most of the forecasting models, making automated deep learning for time series forecasting substantially faster than before.

Evaluating an architecture could take lots of resources. One-shot NAS approaches alleviate this by constructing a supernet where the weights of all the candidate architectures are inherited from this super network. However, training a supernet still requires lots of resources. Zero-Cost (ZC) proxies (Chen et al., 2022; Krishnakumar et al., 2022) attempt to address this problem by estimating the performance of the target model without updating the model weights, thereby greatly accelerating the search process. However, nearly all existing zero-cost proxies (Abdelfattah et al., 2021; Lee et al., 2019; Lin et al., 2021; Ning et al., 2021; Tanaka et al., 2020; Turner, 2019; Wang et al., 2019) are developed for computer vision NAS tasks that only contain one architecture type (convolutional operations). It is still unclear if the ZC proxies can be generalized to compare the performances between different operation types.

## 3 Problem Setting

Time series forecasting tasks aim to predict the values of the target variables for a number of iterations after a certain time step, with the observed same variables and several other feature variables. Formally, given a dataset that is composed of multiple series $\mathcal{D} = \{\mathcal{D}_i\}_{i=1}^N$, where each series is composed of the past observed targets $\mathbf{y}_{i,1:T_i}$, past observed features $\mathbf{x}_{i,1:T_i}$, and known future features $\hat{\mathbf{x}}_{i,T_i+1:T_i+H}$. Given a required forecasting horizon $H$, the model is asked to predict the target variables $\hat{\mathbf{y}}_{i,T_i+1:T_i+H}$ with all the available information:

$$\hat{\mathbf{y}}_{i,T_i+1:T_i+H} = f(\mathbf{y}_{i,1:T_i}, \mathbf{x}_{i,1:T_i}, \hat{\mathbf{x}}_{i,T_i+1:T_i+H}; \theta) \tag{1}$$

Two model families can mainly handle multi-horizon forecasting tasks: the autoregressive approach and the non-autoregressive approach. Autoregressive approaches (Box et al., 2015; Salinas et al., 2020) only predict one step within one forward pass and iteratively use the predicted value as the known feature that can be further fed to the model. On the other hand, non-autoregressive models (Nie et al., 2023; Zeng et al., 2023; Zhou et al., 2021) directly generate multiple forecasting values within one forward pass. In this work, we mainly focus on searching for non-autoregressive architectures.

Neural architecture search aims at finding the optimal architecture $\alpha_*$ for a given task:

$$\min_{\alpha} \mathcal{L}_{val}(\theta^*, \alpha) \quad \text{s.t.} \quad \theta^* \in \arg\min_{\theta} \mathcal{L}_{train}(\theta, \alpha) \tag{2}$$

The search space in previous NAS work still focused on the homogeneous search space, where all the operations belong to the same architecture families. This work presents a unified heterogeneous search space containing most of the potential architectures applied for time series forecasting tasks.

# 4 Search Space Design

It is a common observation and a foundational assumption of NAS that no single architecture family always outperforms the others; based on the results of Deng et al. (2022), we believe that this also holds for time series forecasting tasks. We are unlikely to define a single type of model that works for all datasets. Additionally, some of the architecture might be dependent on the other decision choices. For instance, if we decide to have an encoder-only *Seq Net*, there is no need for us to search for a *Seq* decoder. Here, we propose a hierarchical search space that incorporates most of the forecasting architecture families described in Section 2.1. We will start from the most basic operation level and gradually decrease the granularity until our search space contains all the required components.

## 4.1 Operation Level

The first level, the operation level, describes the operations that can be used in the network. As shown in Figure 1, our search space follows the DARTS (Liu et al., 2019a) search space, a cell-based architecture where each cell is a directed acyclic graph that contains $N$ nodes, including $N_{in}$ input nodes. Each node represents a latent feature map, and the edges that connect the nodes are the operations applied to the latent feature maps. DARTS defines two types of cells: normal cells and reduction cells. These two cell types share the same form of input feature maps. Therefore, they can be easily concatenated to form an architecture. However, this is not the case for forecasting tasks. *Seq Net* and *Flat Net* transform the input features differently. We cannot easily connect a *Seq* cell after a *Flat* cell and vice versa. Hence, we provide a search space for each of the model families.

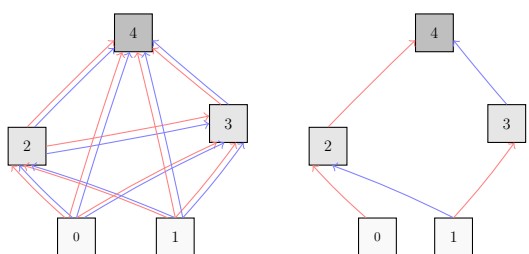

Figure 1: Operation level Search Space. The nodes 0 and 1 are input nodes that receive the network inputs or the outputs from the other cells. Node 4 is the output node. Each colored edge represents an operation. The Search space (Left) is a fully connected directed acyclic graph. Once we have finished the search, we get the final architecture (Right).

Detailed information about each operation can be found in the appendix A.

For the *Seq Net* families, we consider the following operations: 1. MLPMixer (Chen et al., 2023), an all-MLP architecture that applies a linear layer to feature and time dimensions, respectively, 2. LSTM (Hochreiter and Schmidhuber, 1997), 3. GRU (Cho et al., 2014), 4. Transformer (Vaswani et al., 2017), 5. TCN (Bai et al., 2018), 6. SepTCN (Luo and Wang, 2024), and 7. skip connections. These operations are spread to encoder and decoder architectures, which will be discussed in the following section.

For the *Flat Net* families, we have: 1. Linear (Zeng et al., 2023), a single linear layer that encodes the past information to the future variable, 2. N-BEATS (Oreshkin et al., 2020), a repeatedly stacked MLP block, where each block contains a set of fully connected (FC) backbone layers, a forecast, and a backcast head, and 3. skip connections. There are several variants in N-BEATS modules: generic model, trend model, and seasonality model. We incorporate all these modules into our search space, providing another two operations. Since the only difference between different N-BEATS modules is their forecast and backcast head, we ask these N-BEATS modules to share the same FC layers backbones. In total, we have five operations for each edge in the *Flat Net* cell.

Unlike the previous search space that focused on one single architecture type (Klyuchnikov et al., 2020; Krishnakumar et al., 2022; Mehta et al., 2022; Ying et al., 2019), the operations contained in DARTS-TS come from different families. Therefore, the outputs from each operation might have completely different distributions or even different output formats for the same input. The zero-cost proxies that are proven to

be efficient on other benchmarks, such as *flops* or *number of parameters* (Krishnakumar et al., 2022; Wan et al., 2022), might no longer work well within this search space.

## 4.2  Micro Network Level

*Flat* operations only receives the past information $\mathbf{y}_{i,1:T_i}$. Therefore, we stack several *Flat* cells as a *Flat Net*. As shown in Figure 2a, the past targets are first transposed and then fed to the encoder layers. The transposed target, i.e., the backcast part, and a zero tensor whose length is equal to the forecasting horizon, i.e., the forecast part, are fed to the *Flat* encoders. Finally, the forecast output is fed to the forecasting head to predict the target values.[3]

Different from the *Flat Net*, we decompose the *Seq* architectures into two parts: encoders and decoders. The encoders encode the past observed values into an embedding and feed them to the decoder networks. The design of the *Seq* encoder is similar to the *Flat* Encoder, and we stack the encoder cells to form the encoder network. However, as discussed in Section 2.1, two potential ways exist to transform the encoder latent features to the forecasting heads. Therefore, we design the following two types of decoder networks: *Seq* decoder and *Flat* decoder for *Seq* encoder.

The design of the *Seq* Encoder and *Seq* Decoder architecture has been widely applied in previous architecture works (Lim et al., 2021; Sutskever et al., 2014; Vaswani et al., 2017). However, architecture decoders might require different information from the encoder network. For instance, a Transformer decoder (Vaswani et al., 2017) only requires the output from the last layer of the corresponding encoder; an RNN decoder would need the hidden states from the corresponding encoder layer. We record the following information stored by each edge:

1. The output hidden feature map of this edge. It will then be contacted with the corresponding decoder feature maps and fed to the TCN decoder network.

2. The last step's feature map. This value is considered a hidden state that can be fed to the corresponding GRU and LSTM layers to initialize their states.

3. The cell gate state that is applied to initialize the hidden cell states of the corresponding LSTM layer. If the encoder is not an LSTM network, following the idea of stitchable network (Pan et al., 2023) that uses a linear layer to stitch two networks with different shapes, we use a linear layer that transforms the hidden states into the cell gate states.

Hence, we record all the related information as intermediate states during the forward pass. This information is then used to inform the decoder networks of the information provided by the encoder.

On the other hand, we might not want a complex decoder architecture since the information provided to the decoder and most of the features presented in the past are no longer available in the future. Hence, here we define another type of decoder for *Seq* network: the linear (or flat) decoder for *Seq Net*. We apply one linear layer across the time series dimension (Chen et al., 2023; Zeng et al., 2023) that maps the encoder output and the available future information to the decoder output feature. This feature is then fed to the forecasting head to generate the final prediction result. The overall *Seq Net* architecture is presented in Figure 2b.

We also consider the choice of the *Seq Net* decoder as part of the architecture search procedure. Hence, we assign another set of architecture parameters to the output of the two decoder architectures. This architecture is jointly optimized with the other operations architecture weights introduced in Section 4.1.

## 4.3  Macro Architecture Level

The *Flat Net* and *Seq Net* families defined above encode the input data from different perspectives: *Flat Net* families encode the input series along the time series dimension, while *Seq Net* families encode the input series

---

[3]We note that the head here does not necessarily need to be a network module since *Flat* net only predicts one variable each time.

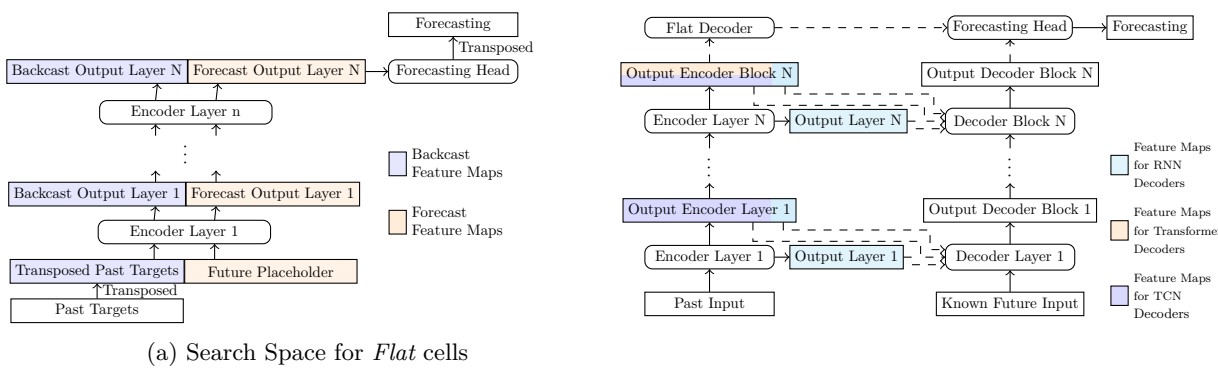

(a) Search Space for *Flat* cells

(b) Search Space for *Seq* cells

Figure 2: Micro-level search space design for Flat and Seq cells.

across different variations. Hence, the two architectures can complement each other, and we concatenate the two architectures sequentially.

As shown in Figure 3, the past target values are first fed to the *Flat Net*, which results in a backcast and forecasting feature maps. The forecast feature maps are then concatenated with the known future features and fed to the *Seq* decoder. Finally, the final forecasting result is the weighted sum of both *Flat Net* and *Seq Net*[4]:

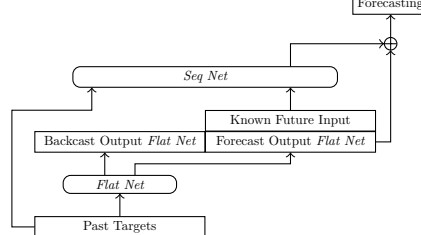

Figure 3: Marco Search Space.

$$f_{model}(\mathbf{y}_{i,1:T_i}) = w_{seq} f_{seq}(\mathbf{y}_{i,1:T_i}, f_{flat}(\mathbf{y}_{i,1:T_i})) + w_{flat} f_{flat}(\mathbf{y}_{i,1:T_i}) \tag{3}$$

These weights are considered an architecture parameter that can be jointly learned with the other architecture parameters described in the aforementioned sections.

## 5 Searching for the Optimal Architectures

Here, we provide an exemplary searching strategy, i.e., DARTS (Liu et al., 2019b; Wang et al., 2021), to search within our search space. DARTS assigns a weight for each of the operations within its search space and optimizes these architecture weights jointly with the model weights using gradient descent. Some of the modules, such as Dropout (Srivastava et al., 2014) and Batch Normalization (Ioffe and Szegedy, 2015), behave differently during training and inference time. We thus switch these modules to the *eval* mode when we update the architecture weights with validation losses to simulate the evaluation process.

During the search phase, we divide the official training/validation splits of the raw dataset into training and validation sets of the same size, such that they do not overlap with the test sets. The training set and validation sets are then used to optimize the architecture weights and architecture parameters, respectively. We follow the common practice for the training-validation split in time series forecasting tasks: the validation set is located at the tail of the training set. The first half of the dataset is considered the training set that is used to optimize the weights of the supernet, while the second part of the dataset is the validation set, which is used to optimize the architecture parameters. We apply RevIN (Kim et al., 2022) during both the searching and training phases to ensure that the input features fed to the networks stay in the same distribution.

---

[4]For the sake of simplicity, we omit the feature variables $\mathbf{x}_{i,1:T_i}$ and $\hat{\mathbf{x}}_{i,T_i+1:T_i+H}$ that are fed to the $f_{model}$ and $f_{seq}$

| | DARTS-TS | | iTransformer | | ModernTCN | | PatchTST | | TSMixer | | DLinear | | TimesNet | | Autoformer | |
| --- | --- | --- | --- | --- | --- | --- | --- | --- | --- | --- | --- | --- | --- | --- | --- | --- |
| | MSE | MAE | MSE | MAE | MSE | MAE | MSE | MAE | MSE | MAE | MSE | MAE | MSE | MAE | MSE | MAE |
| ETTm1 | 0.344 | 0.368 | 0.368 | 0.395 | 0.362 | 0.386 | 0.352 | 0.381 | 0.382 | 0.408 | 0.360 | 0.382 | 0.402 | 0.415 | 0.619 | 0.539 |
| ETTm2 | 0.253 | 0.306 | 0.273 | 0.330 | 0.261 | 0.319 | 0.257 | 0.315 | 0.446 | 0.477 | 0.267 | 0.329 | 0.290 | 0.339 | 0.423 | 0.441 |
| ETTh1 | 0.413 | 0.423 | 0.473 | 0.468 | 0.404 | 0.421 | 0.415 | 0.429 | 0.515 | 0.503 | 0.447 | 0.456 | 0.486 | 0.482 | 0.559 | 0.529 |
| ETTh2 | 0.351 | 0.388 | 0.387 | 0.415 | 0.333 | 0.385 | 0.330 | 0.379 | 0.571 | 0.548 | 0.422 | 0.439 | 0.399 | 0.435 | 0.739 | 0.621 |
| ECL | 0.156 | 0.245 | 0.166 | 0.261 | 0.163 | 0.257 | 0.161 | 0.254 | 0.168 | 0.272 | 0.166 | 0.264 | 0.203 | 0.302 | 0.221 | 0.334 |
| Exchange | 0.378 | 0.409 | 0.411 | 0.440 | 0.525 | 0.505 | 0.385 | 0.417 | 0.366 | 0.452 | 0.382 | 0.419 | 0.540 | 0.524 | 1.010 | 0.775 |
| Weather | 0.230 | 0.262 | 0.239 | 0.274 | 0.231 | 0.269 | 0.229 | 0.265 | 0.222 | 0.288 | 0.244 | 0.297 | 0.249 | 0.287 | 0.398 | 0.431 |
| Traffic | 0.394 | 0.260 | 0.387 | 0.273 | 0.421 | 0.287 | 0.398 | 0.267 | 0.541 | 0.413 | 0.434 | 0.295 | 0.624 | 0.336 | 0.671 | 0.412 |

Table 1: The results of the mean performance over all the long-term forecasting datasets. The best models are marked in red, while the second-best model is marked with underlines.

| | DARTS-TS | | iTransformer | | ModernTCN | | PatchTST | | TSMixer | | DLinear | | TimesNet | | Autoformer | |
| --- | --- | --- | --- | --- | --- | --- | --- | --- | --- | --- | --- | --- | --- | --- | --- | --- |
| | MSE | MAE | MSE | MAE | MSE | MAE | MSE | MAE | MSE | MAE | MSE | MAE | MSE | MAE | MSE | MAE |
| PEMS03 | 0.137 | 0.241 | 0.458 | 0.408 | 0.408 | 0.419 | 0.199 | 0.291 | 0.162 | 0.281 | 0.264 | 0.358 | 0.151 | 0.248 | 0.554 | 0.541 |
| PEMS04 | 0.112 | 0.221 | 0.127 | 0.237 | 0.488 | 0.471 | 0.266 | 0.338 | 0.136 | 0.254 | 0.264 | 0.355 | 0.127 | 0.239 | 0.763 | 0.669 |
| PEMS07 | 0.102 | 0.204 | 0.504 | 0.477 | 0.304 | 0.374 | 0.209 | 0.293 | 0.150 | 0.251 | 0.311 | 0.373 | 0.132 | 0.233 | 0.361 | 0.438 |
| PEMS08 | 0.169 | 0.256 | 0.202 | 0.269 | 0.510 | 0.479 | 0.231 | 0.304 | 0.225 | 0.305 | 0.331 | 0.376 | 0.191 | 0.267 | 0.739 | 0.627 |

Table 2: The results of the mean performance over all the PEMS datasets. The best models are marked in red, while the second-best model is marked with underlines.

## 5.1 Hierarchical Pruning of the One-Shot Model

Vanilla DARTS are shown to be unstable during the search process and might prefer to select architectures that are dominated by skip connections (Chu et al., 2020; Jiang et al., 2023; Wang et al., 2021; Zela et al., 2020). Hence, once the weights and architecture parameters are trained, as the last step, we select the optimal operations and edges using the perturbation-based approach (Wang et al., 2021).

Given that our search space is a hierarchical search space, some of the operations might be dependent on others, and we have to consider that for the pruning phase. For instance, if we select a Linear Decoder, then we do not need to further select the operations in the *Seq* decoder. However, on the other hand, this also indicates that our estimate will be biased if we select the choice of decoder before selecting any edge operations in the *Seq* Decoder. Hence, we propose pruning our network from the lowest granularity level and gradually increasing the granularity level until we prune the operations in our search space. Once all the operations are selected, we further prune the edges of our network using the same perturbation-based approach. Finally, only two edges are preserved for each node, including the cell output node.[5]

## 6 Experiments

In this section, we first demonstrate that DARTS-TS can identify the optimal architecture, which is comparable to many other handcrafted architectures on various datasets. We then provide a brief analysis of our search space to show the potential challenges in our search space. After that, we provide an analysis of the latency of the optimal architecture. Finally, we show some optimal architectures as examples.

### 6.1 Time series forecasting tasks

We evaluate our architecture search framework on the popular long-term forecasting datasets introduced by Wu et al. (2021) and Zeng et al. (2023): Weather, Traffic, Exchange Rate (Exchange), Electricity (ECL), and four ETT datasets. Additionally, we evaluate our approach on the four PEMS datasets (Chen et al., 2001) that record the public traffic network data in California.

Following the experiment setup from the other works, we set the forecasting horizon $H$ for ETT, ECL, Exchange Rate, Weather, and Traffic datasets as $\{96, 192, 336, 720\}$. For the four PEMS datasets, we set

---

[5]We note that this setting is different from the traditional NAS framework, where all the edges towards the output nodes are preserved. This approach helps us to reduce the architecture size and required latency further.

these values as $\{12, 24, 48, 96\}$. We compare our results with the following baselines: PathTST (Nie et al., 2023), ModernTCN (Luo and Wang, 2024), DLinear (Zeng et al., 2023), TSMixer (Chen et al., 2023). iTransformer (Liu et al., 2024b), Autoformer (Wu et al., 2021) and TimesNet (Wu et al., 2023). For the sake of fair comparison, we follow the setup from PatchTST (Nie et al., 2023) and set the input sequence length for all models to 336. The training/validation/test splits of each dataset follow the rules defined in the Time Series Library[6], where a fixed ratio of the series is described as test sets, i.e., each test set might contain more time steps than the required forecasting horizons. i.e., we perform rolling origin evaluation by continuously moving the forecasting origin until we iterate through the entire test set. We ran each experiment 5 times with different seeds and recorded their mean and standard deviation, respectively.[7] Further experimental details, including the datasets information, searching and evaluation costs, hyperparameter settings, and GFlops changes after each stage, can be found under the appendix B.

The results for long-term forecasting tasks are shown in Table 1. The best results are marked in red. The full results can be found in the appendix C, where we also provide a statistical analysis against the best baseline models on each dataset. Our network achieves the best or comparable results on the ECL, ETTm, Traffic, and Weather Datasets. Overall, we show that DARTS-TS automatically found architectures that are comparable to or better than many other hand-crafted architectures specifically designed for forecasting tasks with only the vanilla series modules.

While on another benchmark, the PEMS dataset, DARTS-TS outperforms all the other baselines for all the tasks, as shown in Table 2. This shows that DARTS-TS could adapt to different tasks and suggests optimal architectures for different time series distributions.

## 6.2 Search Space Analysis

Unlike the search space in other tasks, which contains only a single operation type (Klyuchnikov et al., 2020; Krishnakumar et al., 2022; Mehta et al., 2022; Ying et al., 2019), our search space encompasses operations from different architectures. This differs from the previous architecture search space and may present additional challenges to the optimizers. To provide a basic understanding of our search space, we randomly sample $3,000$ configurations from our search space and evaluate them on the ECL and Traffic datasets.

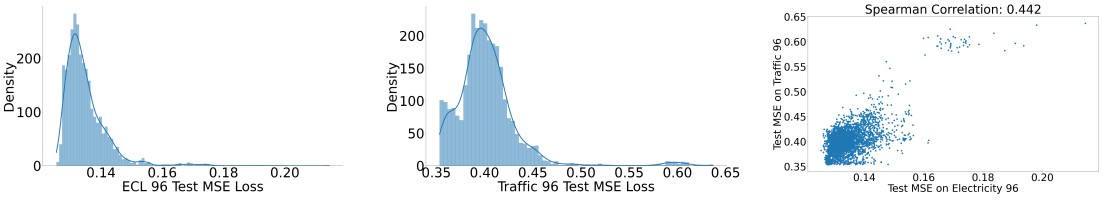

Figure 4: Random Search Evaluation Results on the ECL and Traffic datasets. (Left), MSE loss distributions on the ECL dataset. (Middle), MSE loss distributions on the Traffic dataset. (Right), loss distributions on the Traffic ECL dataset from the same architecture configurations

The random configuration performances are shown in Figure 4. For the same set of hyperparameter configurations, the MSE losses on these two tasks follow different distributions: losses on the ECL datasets are more skewed towards the lower bounds, while the losses on the Traffic datasets are more uniformly distributed. This indicates different difficulty levels for the optimizers to search for the optimal architectures on different datasets. To check whether the performance of the same hyperparameter configurations could transfer to different tasks, we plot the performance of the same configurations from the two tasks in the right part of Figure 4. The overall trend shows the consistent performance of the worst-performing configurations on both datasets. However, the performance diverges as we move towards the near-optimal configurations. Hence, the optimal configuration for one task may not work equally well for another task. There is still a need to further search for the optimal model on the target dataset. However, since the worst configurations

---

[6]https://github.com/thuml/Time-Series-Library
[7]Our code can be found on https://github.com/automl/OneShotForecastingNAS.git.

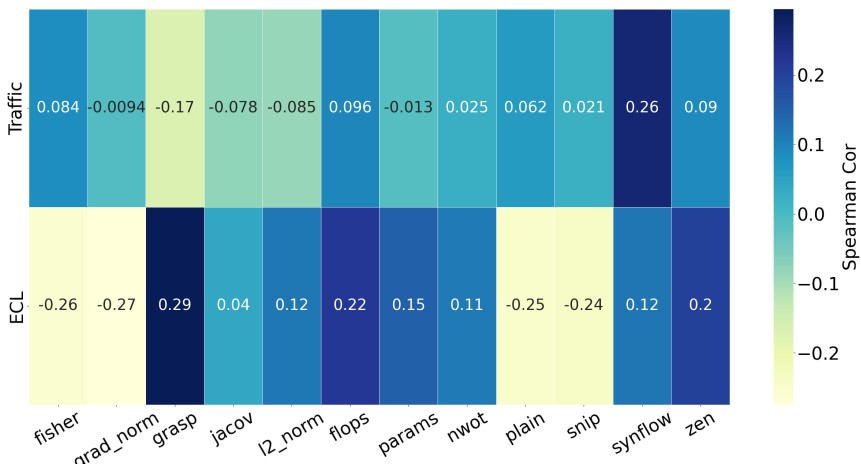

Figure 5: Spearman correlation between different ZC metrics and the evaluation test MSE losses

perform consistently on the two tasks, we could incorporate the runs from other tasks as a prior (Hvarfner et al., 2022; Mallik et al., 2023).

Operations in our search space come from different architecture types. This provides further challenge for the optimizers as the performance difference among the operations in the search space can no longer be described with simple proxies. For instance, in the computer vision NAS benchmark search space, a $3 \times 3$ convolutional layer could, in most cases, achieve a better performance than a $1 \times 1$ layer, as the *number of parameters* of $3 \times 3$ convolutional layer is larger than that of $1 \times 1$ convolutional layer. However, it is questionable if a Transformer layer (with $12d^2$ parameters) will outperform an LSTM layer (with $8d^2$ parameters) since these two operations follow different computational rules. Hence, the widely applied zero-cost proxies applied in the computer vision NAS benchmarks might no longer work in our search space.

We computed the zero-cost (ZC) metrics for the sampled architectures. Following the settings from NAS-Bench-Suite-Zero (Krishnakumar et al., 2022), we evaluate the following zero cost proxies: fisher (Turner, 2019), flops, number of parameters (Ning et al., 2021), grad-norm, l2-norm, plain (Abdelfattah et al., 2021), grasp (Wang et al., 2019), jacov, nwot (Mellor et al., 2020), snip (Lee et al., 2019), synflow (Tanaka et al., 2020), and zen-score (Lin et al., 2021). We evaluate the ZC scores of all the sampled models and compute the Spearman correlation between the ZC scores and their test performance. The results are shown in Figure 5. Supervisingly, even though many zero-cost proxies do not limit their application to pure CNN or image classification tasks, nearly all of the zero-cost proxies fail in our search space. This indicates that the existing zero-cost proxies might not generalize well to more complex search spaces. Additionally, as shown in Figure 4, even the same configuration might perform differently on different datasets. This further shows the necessity of defining new data-dependent zero-cost proxies that work well across different architecture types.

To provide further insight into the design of ZC proxies for our search space, we check the performance of the architectures that contain at least one operation within our search space (Lopes et al., 2023). This provides a preliminary estimation of the strength of each operation on the architecture performance. The result is shown in Figure 6. Although the sequential network with sequential decoder families performs similarly to the sequential network with linear decoder on the ECL dataset, the gap becomes larger when the same architecture is evaluated on the Traffic dataset. However, the linear decoders might also lead to a much worse model, as the worst-performing linear decoders' loss is much higher than that of the sequential decoders.

Among the sequence operations, the TCN families (TCN and separated TCN models) achieve better median and 25th quantile performance on both datasets, which indicates that the dataset requires the models to focus more on local dependencies. While transformer families dominate the other benchmark, their median and 25th quantile losses are higher than those of the other operations. Although the MLP mixer encoders are quite close (and sometimes are even optimal) to the other operations, MLP Mixer decoders perform

worse than the other sequential decoding operations. This highlights the importance of incorporating various encoder-decoder architectures into forecasting architecture designs.

For flat operations, MLP Flat and N-BEATS-Generic achieve better performance compared to N-BEATS-Seasonal and N-BEATS-Trend models. However, the top-performing N-BEATS-Seasonal models achieve a lower loss compared to the top-performing MLP flat layers. This might indicate that the inductive bias contained in the N-BEATS-Seasonal models, i.e., the forecasting results are periodic, would help the model achieve better performance.

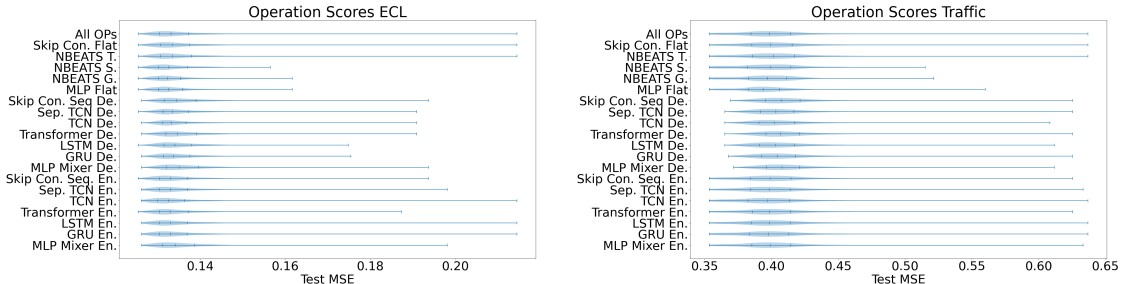

Figure 6: Performance of the randomly sampled models that contain at least one operation

We further show the scores of the architectures that do not contain the target operations in Figure 7. In this case, a higher score indicates that the target function is more important for the target task. Results show that many operations have different impacts on the searched architectures. For instance, missing N-BEATS-Trend results in a poorer optimal architecture for the ECL dataset, while a missing separated TCN results in a poorer mean-performing architecture on the same dataset. For the traffic dataset, removing the same operation could even yield a better-performing set of architectures. This shows that even the same operation might have a completely different influence on the architectures in the search space.

## 6.3 Model Efficiency Analysis

The growing demand for forecasting models has posed more challenges to the forecasting networks: the network should be fast, so that it can quickly predict the following trend. Additionally, networks need to contain fewer parameters and consume less memory so we can deploy them on embedded systems.

To further show that our network could find an efficient and strong network, we ask all the networks to do a single forward pass and backpropagation with the series within the Traffic and ECL dataset, where each series contains 862 and 321 variables, respectively. We set the batch size of the series to 32 and the look-back window size to 96. The networks are then asked to predict the future series with a forecasting horizon of 96. This experiment is executed on one single Nvidia 2080 TI GPU with 11 GB GPU RAM[8]. Due to this

---

[8]However, we search the one-shot model on an Nvidia A100 GPU with 40 GB GPU RAM.

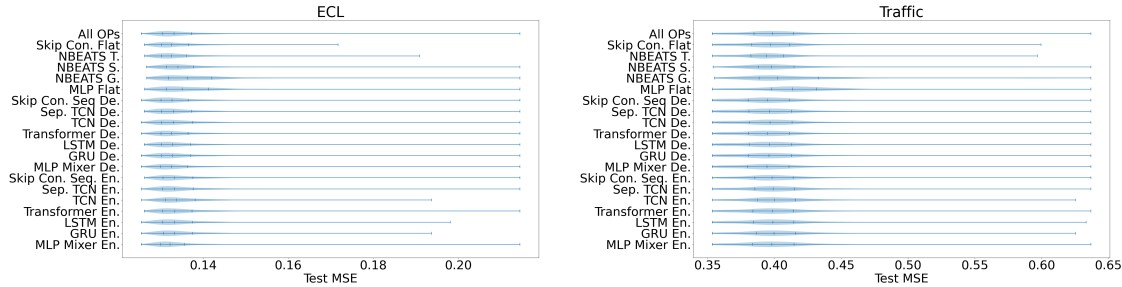

Figure 7: Performance of the randomly sampled models that do not contain the specific operation

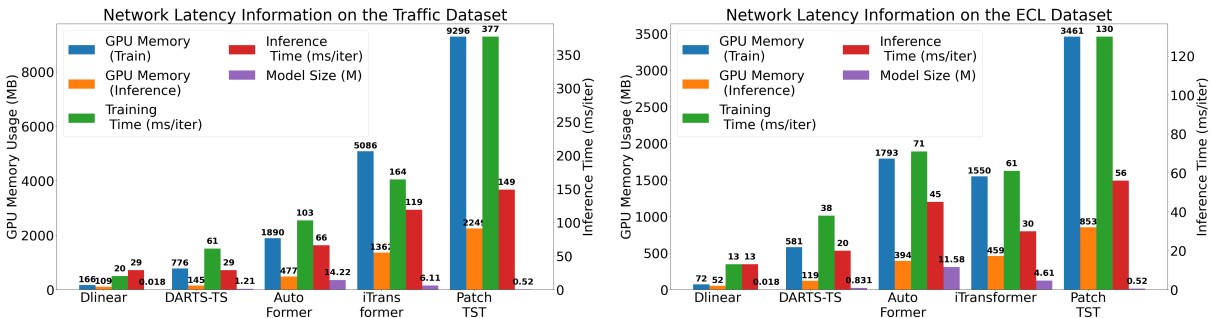

Figure 8: Latency information of different networks on the Traffic (left) and ECL (right) dataset. We set the look-back window size and the forecast horizon to 96. The batch size for both training and inference is set as 32.

memory constraint, some of the networks, such as ModernTCN, cannot fit into this GPU with our setup and do not appear in this comparison.

As shown in Figure 8, while having a comparable performance with iTransformer and PatchTST on the traffic dataset, our approach requires around 2x less GPU memory and is faster than the iTransformer and 3x less GPU memory and speed up compared to PatchTST during the training phases. During the test phases, the required GPU memory is even further reduced to around 300 MB, which is 4x less than the iTransformer and 6x less than the PatchTST, and only requires 3x more memory compared to a linear layer. A similar trend can be observed on the ECL dataset: DARTS-TS requires 7x less GPU memory and is 2.8x faster than PatchTST to achieve a better performance.

Although we do not optimize for the latency as our optimization objects explicitly, the combination of different operations already allows us to achieve a similar performance with much less computational power required. e.g., to cover the same receptive field, we might need to stack multiple TCN layers or increase their kernel size. Both approaches result in an increased amount of computational time and memory consumption. However, if we instead apply a transformer or RNN layer on top of a TCN layer, the model could still learn the global and local information with less computational power required. Additionally, models such as PatchTST (Nie et al., 2023) decompose one single multi-variant series instance into multiple uni-variant series instances and therefore, need to run the forward pass multiple times for a single multi-variant time series instance. DARTS-TS only performs one forward pass to get the forecasting results, which could significantly reduce the required computational power and the memory requirements. Hence, DARTS-TS could search for a more lightweight architecture, even if latency information is not the optimization target. However, we could also apply other optimization approaches, such as multi-objective differentiable architecture search (Sukthanker et al., 2024), to search for the architecture that concerns both latency and accuracy.

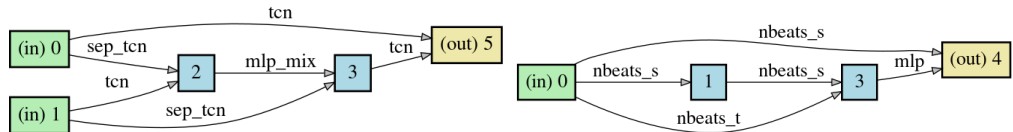

Figure 9: One optimal architecture on the ECL dataset, this is an encoder-only architecture and will be trained with an MSE loss

## 6.4 The optimal architectures

We show one of the optimal architectures found on the ECL dataset in Figure 9. Its *Seq Net* is an encoder-only architecture that is composed of MLPMixer, TCN, and Separate TCN modules. TCN modules are still preferable over the other components, showing that the ECL dataset might prefer a model that focuses on the

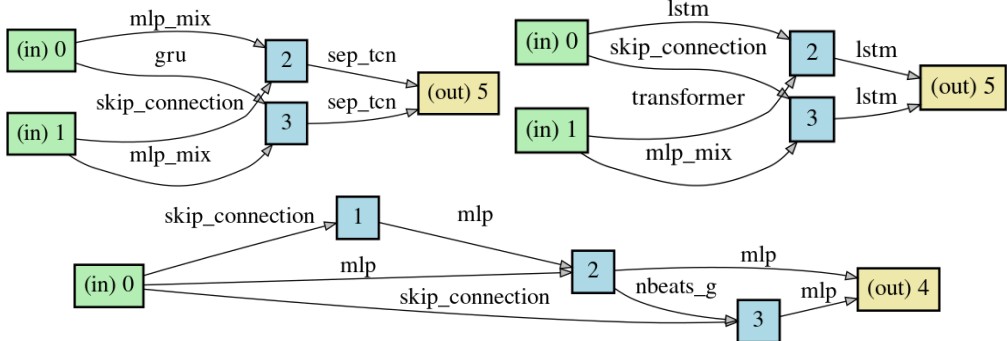

Figure 10: One optimal architecture on the ETTm2 dataset, this is an encoder-decoder architecture and will be trained with a quantile loss

local correlation. While the *Flat Net* contains lots of N-BEATS-Seasonal modules, this indicates the strong seasonal and little trend signal that the dataset contains.

We present another encoder-decoder architecture in Figure 10. This architecture is optimized on the ETTm2 dataset. It applies two MLP-Mixer layers to the input node to first collect the global information from the raw input sequence and then apply two Separate TCN modules on top of that. Our optimizer also selects many LSTM modules for the decoder architectures, even if the corresponding encoder edges do not provide any hidden states. Additionally, no TCN module in the decoder layer. This is different from the optimal encoder architecture, where lots of TCN family components are selected. This indicates that the decoder networks would require modules that provide a global perspective to utilize all the information from the encoder networks, and therefore, the priority of TCN modules might decrease. More optimized architectures can be found in the appendix F.

## 7 Discussion and Future Work

This work proposes a general search space for time series forecasting tasks. Our search space allows the components in the search space to freely connect to each other and form a new network. This search space contains most of the forecasting architectures and can be easily extended to other frameworks. For instance, iTransformer (Liu et al., 2024b) can be considered as a special case of the *Flat Net* and searched jointly with the other modules from this family. Decomposing multi-variant series into single variant series and applying a special kernel result in PatchTST (Nie et al., 2023), and then we can search for the optimal architecture jointly with the other *Seq Net*.

In Section 6, we showed that DARTS-TS can search for a lightweight architecture while keeping strong performance on various datasets. However, unlike traditional supervised problems where all the sample instances are i.i.d., time series data might have the problem of distribution shift. The optimal model searched on the validation set might no longer work well on the test set. This provides a future challenge for the AutoML forecasting frameworks from the meta-level, i.e., they need to be able to pre-determine the optimal approach to evaluate the generalization ability.

## 8 Acknowledgement

The authors gratefully acknowledge the computing time provided to them on the high-performance computers Noctua2 at the NHR Center PC2 under the project hpc-prf-intexml. These are funded by the Federal Ministry of Education and Research and the state governments participating on the basis of the resolutions of the GWK for the national high performance computing at universities (www.nhr-verein.de/unsere-partner).

Difan Deng was supported by the Federal Ministry of Education and Research (BMBF) under the project AI service center KISSKI (grant no.01IS22093C).

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

# A    Operations Deatails

In section  4.1, we briefly introduced the operations within our search space. Here, we will provide the details of these operations.

## A.1    *Seq Net*

For *Seq* net, encoders and decoders share the same operation sets. Overall, we have the following operations:

- TSMixer (Chen et al., 2023), a full MLP-based Sequential operation. Each TSMixer operation is constructed by the time and feature mixing blocks. The time mixing blocks use a fully connected (FC) layer to mix the information across different time steps. In contrast, the feature mixing block uses another set of FC layers to enhance the information within each channel. Our TSMixer encoders follow the design from  Chen et al. (2023), and the size of the feature mixing layer is set as $2 \times d_{model}$. For TSMixer decoders, we first construct the input feature map with the encoder network outputs. This concatenated feature is then provided to the time mixing modules to recover its size and return it to the forecasting horizon. Additionally, we use LayerNorm (Ba et al., 2016) instead of BatchNorm (Ioffe and Szegedy, 2015) in TSMixer to ensure that the operations within an edge generate the feature maps that follow the same distribution (since all other components in our module used LayerNorm to normalize the feature maps).

- LSTM (Hochreiter and Schmidhuber, 1997) is an RNN model. It maintains a set of cell gate states to control the amount of information passed to the next time steps. Therefore, it suffers less from the known gradient explosion problems in the RNN families. However, the introduction of the gates brings lots of additional parameters to the modules. As described in Section 4.2, the hidden states of the LSTM decoder are initialized by the last time step of the encoder feature from the corresponding layer and another feature generated with a linear embedding (or directly from the corresponding LSTM encoder).

- GRU (Chung et al., 2015) is yet another type of RNN network. It only maintains one hidden state and therefore requires a much smaller number of parameters and computations compared to the LSTM. The setting of the GRU Encoder/Decoder is nearly the same as the LSTM families. The only difference is that we do not maintain an additional state to initialize the GRU decoders.

- Transformer (Vaswani et al., 2017) has attracted lots of attention from different research fields and is therefore widely applied in time series forecasting tasks. Here, we use the vanilla Transformer implemented in PyTorch (Paszke et al., 2019) with the hidden size to be $4 \times d_{model}$.

- TCN (Bai et al., 2018) is a type of CNN network that can capture the local correlations among different time steps. However, the receptive field of the TCN network is restricted by its kernel size. To efficiently increase the receptive field without introducing too much computation overhead with a larger kernel, TCN implemented dilated convolution operations. Here we implement a similar approach, for edge $i \leftarrow j$ that starts from node $j$ to node $i$ as cell $k$, we set its dilation as $2^{(j+k-n_{in})}$, where $n_{in}$ is the number of inputs of the current cell. Hence, the deeper convolutional layers will have a larger receptive field. For the TCN decoders, we concatenate the feature maps from the corresponding encoder layer with our input feature and feed them together to our TCN network. This idea is similar to U-Net (Ronneberger et al., 2015), where features with similar levels should be gathered together.

- SepTCN (Luo and Wang, 2024) is a variation of the vanilla TCN. We replace the full convolutional operations in TCN with a combination of a separated TCN model and another $1 \times 1$ linear layer.

- Skip Connection, an identity layer that passes its input to the next level. However, for the skip connection encoder, we still have a linear layer to provide initial cell gate states to the corresponding LSTM decoder layers.

Another type of *Seq* decoder is a linear decoder. We apply a linear layer that transforms the encoder's output feature map with size $R^{[B,L,N]}$ to $R^{[B,H,N]}$ and feed it further to the forecasting heads.

## A.2  *Flat Net*

We only consider the MLP families in our *Flat Net* to minimize the computational overhead when applying our approaches to problems with higher series amounts. Given an input feature series with shape $[B, N, L]$, we first concatenate it with a zero tensor with shape $[B, N, H]$ that represents the prediction results. Then this concatenated tensor is fed to the *Flat* encoder.

This architecture family includes:

- a simple Linear model (Zeng et al., 2023). This linear layer maps its input features with shape $[B, N, L + H]$ to a feature map whose size is equal to the forecasting horizon: $[B, N, H]$. Then the output feature is concatenated with the first part of the input feature maps. If the network only contains skip connections for all but the last layer, then this network becomes a DLinear model (Zeng et al., 2023). If the operation is not the output layer, we attach an activation and normalization layer to introduce some non-linearity.

- N-BEATS (Oreshkin et al., 2020) modules. N-BEATS is a hierarchical module where each model is composed of multiple stacks. Each stack contains multiple blocks. Each block has an FC stack with multiple FC layers, a forecasting head, and a backcasting head. In our search space, each N-BEATS edge corresponds to an N-BEATS block. N-BEATS provides three variations: generic, trend, and seasonal. We include them all in our search space. Since the only difference between these variations is their prediction heads. We ask the models to share the same FC layer backbones and only diverge at the forecasting heads.

- Skip Connection, a skip connection layer.

## A.3  Forecasting Heads

We also consider the forecasting heads as part of the operations within our graph. Each of these heads is composed of one or multiple linear layers that map the *Seq Net* [9] output feature maps to the desired multiple-variable target values. These linear layers are then trained with a set of specific training losses. Let's assume that the target value is $\mathbf{y}$ and prediction value is $\hat{\mathbf{y}}$

- Quantile loss (Lim et al., 2021; Wen et al., 2017) predicts the percentiles of the target values. A quantile head can be composed of multiple heads, and each of the heads is asked to predict a $q$ quantile. Given a required quantile value $q$, the quantile loss is computed by $\mathcal{L}_q = \max(q(\mathbf{y}-\hat{\mathbf{y}})+(1-q)(\hat{\mathbf{y}}-\mathbf{y}))$. In our network, we used the following quantile values: $\{0.1, 0.5, 0.9\}$. The final prediction is given by the 0.5 quantile values. A quantile head is then a set of linear layers whose size is the number of quantile values

- MSE loss is yet another popular choice in time series forecasting tasks. It is computed by $\mathcal{L}_{MSE} = (\hat{\mathbf{y}} - \mathbf{y})^2$. An MSE head is a single linear layer whose weights are updated with MSE loss.

- MAE loss is similar to MSE loss. However, instead of computing the mean square error from MSE, it computes the mean absolute error: $\mathcal{L}_{MAE} = |\hat{\mathbf{y}} - \mathbf{y}|$. Similar to the MSE layer, an MAE head is also composed of one linear layer, but its weights are updated with MAE losses.

We stack these forecasting heads on top of the *Seq Net* decoders and optimize their architecture weights with validation losses.

---

[9]For *Flat Net*, there is no need to have an additional head if the loss only requires one output

# B  Experiment Details

We show detailed information on the dataset that we applied in Table 4. Given the great discrepancy in variable size between different datasets, we divide the datasets into two groups: we assign a smaller model to the datasets with fewer variables, such as Weather, Exchange, and ETTs. We then attach a normalization layer within the linear decoder. For the other datasets, we design an architecture search space with a relatively large model. Additionally, we removed the normalization layer in the linear decoder since the number of target variables is larger than our model size, and the final forecasting head does not need to recover the distribution of the target variable from the normalized feature maps.

We run our experiments on a Cluster equipped with Nvidia A100 40 GB GPUs and AMD Milan 7763 CPUs. For each dataset, we perform the search over the smallest forecasting horizons and evaluate the same optimal model on all the other forecasting horizons. Each task is repeated 5 times. The resources spent on each evaluation depend on the size of the dataset. The exact GPU hours spent for each task and stage are presented in Table 3. The search is divided into three stages. The first stage is to jointly optimize the network weights and parameters, and the second stage is to select the optimal operations within each edge. Finally, the third stage is applied to only preserve two edges toward each node. Overall, it takes roughly 4 hours to evaluate smaller datasets such as ETThs and Exchange. Other tasks might require up to 10 GPU hours. The ablation study requires another 500 GPU hours. Overall, it takes roughly 1200 GPU hours to finish the experiments.

|  | ECL | ETTh1 | ETTh2 | ETTm1 | ETTm2 | Traffic | Exchange | Weather |
|---|---|---|---|---|---|---|---|---|
| Optimization | 3.665 | 0.845 | 0.845 | 3.512 | 3.594 | 3.187 | 0.818 | 3.273 |
| Operation Prune | 2.672 | 0.730 | 0.738 | 2.984 | 3.024 | 2.445 | 0.584 | 2.316 |
| Topology Prune | 0.088 | 0.024 | 0.030 | 0.106 | 0.106 | 0.107 | 0.022 | 0.078 |
| Search | 6.424 | 1.599 | 1.612 | 6.602 | 6.724 | 5.738 | 1.424 | 5.666 |

Table 3: GPU hours used for the search stage

For all the baselines, we use their official implementation from PatchTST[10], ModernTCN[11], and TSMixer[12], while for the other baselines, we take the implementation from Time-Series-Library[13].

| Dataset | # Time steps | # Variables | Dataset | # Time steps | # Variables |
|---|---|---|---|---|---|
| ECL | 26304 | 321 | ETTm1 | 69680 | 7 |
| Traffic | 17544 | 862 | ETTm2 | 69680 | 7 |
| Weather | 62696 | 21 | PEMS03 | 26208 | 358 |
| Exchange | 7588 | 9 | PEMS04 | 16992 | 307 |
| ETTh1 | 17420 | 7 | PEMS07 | 28224 | 883 |
| ETTh2 | 17420 | 7 | PEMS08 | 17856 | 170 |

Table 4: Data set information

Our architecture search framework is a two-stage approach. In the first stage, we search for the optimal architecture, while in the second stage, we train the proposed network from scratch. We preserve most of the searching hyperparameters from Liu et al. (2018). However, we apply ADAM instead of SGD during the test phases to optimize the network weights. All the optimizers are applied with CosineAnnealing-WarmRestarts (Loshchilov and Hutter, 2017) that restarts learning rates every 20 epochs. We also apply a smaller learning rate and stronger weight decay to the smaller datasets to avoid overfitting. The concrete hyperparameter settings during search and evaluation phases are presented in Table 5

All the look-back window sizes are set to 336 (for long-term forecasting tasks) and 96 (for PEMS tasks) for both the searching and testing phases. We divide the datasets into two groups based on their number of

---

[10]https://github.com/yuqinie98/PatchTST
[11]https://github.com/luodhhh/ModernTCN
[12]https://github.com/google-research/google-research/tree/master/tsmixer
[13]https://github.com/thuml/Time-Series-Library

| HP Names | Search HP Values | | Evaluation HP Values | |
|---|---|---|---|---|
| | Weights Optimizers | Architecture Optimizers | big datasets | small datasets |
| Epochs | 40 | 40 | 100 | 100 |
| Gradient Clip | 0.1 | 0.1 | 0.1 | 0.1 |
| Type | SGD | Adam | Adam | AdamW |
| Learning Rate | 0.025 | 0.001 | 0.001 | 0.0002 |
| Weight Decay | 0 | 0.001 | 0 | 0.1 |
| Momentum | 0.9 | | | |
| Betas | | (0.5, 0.999) | (0.9, 0.999) | (0.9, 0.999) |

Table 5: Training Hyparameters (HP) during searching and training phases

variables: 1. The PEMSs, traffic, and ECL dataset belongs to the big dataset 2. The remaining datasets, including Weather and ETTs, are small datasets.

Both datasets share nearly the same architecture, which is a mixed network introduced in Section 4.3. The number of *Seq* and *Flat* cells is both set to 2, and they could receive 2 and 1 input variables, respectively. However, for the big dataset, we set the number of *Seq Net* hidden dimensions as 32, while this value for the small dataset is 8. This approach is also applied to the hidden NBATS dimension of *Flat Net*: we set this value as 256 for the big datasets and 96 for the small datasets. The hyperparameters applied during training and validation are listed in Table 5

Since the *Seq* encoder has two input nodes, instead of feeding the raw input value to both input nodes, we decompose the input nodes into trend-cyclical components by using an average moving (Wu et al., 2021; Zeng et al., 2023) to ask the edges in the cell to focus on different levels of information.

| | ECL | ETTh1 | ETTh2 | ETTm1 | ETTm2 | Traffic | Exchange | Weather |
|---|---|---|---|---|---|---|---|---|
| Supernet | 26.241 | 0.750 | 0.750 | 0.750 | 0.750 | 61.841 | 0.778 | 1.149 |
| After OP Prune | 4.991 | 0.078 | 0.079 | 0.061 | 0.068 | 15.237 | 0.087 | 0.158 |
| After Topology Prune | 4.358 | 0.042 | 0.060 | 0.031 | 0.040 | 11.911 | 0.052 | 0.079 |

Table 6: GFLOPS changes after different stages

We now study how FLOPS change within each Prune stage. DARTS-TS search process involves three stages: weights optimization, operation pruning, and topology pruning. We show how the flops change after each stage in Table 6. Most of the FLOPS are substantially reduced after the operation pruning stage, where only one operation is preserved for each edge. This value is further reduced with the Topology Prune stage, where each node only preserves at most two edges as inputs.

## C    Results on all the forecasting horizons

Table  7 and Table  8 show the results w.r.t. each forecasting horizon. While DARTS-TS requires a much smaller amount of resources and parameters, it still shows comparable performances on many datasets.

### C.1    Statstical Test

Table 9 and Table 10 show the statistical test against the best non-NAS baseline for each dataset. For the LTSF datasets, the optimal baseline models differed across different datasets. With a significant level of 0.05, DARTS-TS is significantly better than the baseline models on most of the benchmarks for ECL, ETTm1, and ETTm2 datasets. While for the other benchmarks, DARTS-TS is many times comparable to the best baselines, with the forecasting losses slightly higher or lower than the baselines. Additionally, the optimal baselines differ across different datasets, which also shows that the architectures need to adapt to different tasks, further highlighting the importance of heterogeneous forecasting architectures.

| | | DARTS-TS | | iTransformer | | ModernTCN | | PatchTST | | TSMixer | | DLinear | | TimesNet | | Autoformer | |
|---|---|---|---|---|---|---|---|---|---|---|---|---|---|---|---|---|---|
| | | MSE | MAE | MSE | MAE | MSE | MAE | MSE | MAE | MSE | MAE | MSE | MAE | MSE | MAE | MSE | MAE |
| ECL | 96 | 0.129 (0.00) | 0.217 (0.00) | 0.132 (0.00) | 0.228 (0.00) | 0.135 (0.00) | 0.231 (0.00) | 0.130 (0.00) | 0.223 (0.00) | 0.137 (0.00) | 0.241 (0.00) | 0.140 (0.00) | 0.237 (0.00) | 0.184 (0.00) | 0.287 (0.00) | 0.204 (0.00) | 0.320 (0.01) |
| | 192 | 0.147 (0.00) | 0.234 (0.00) | 0.155 (0.00) | 0.249 (0.00) | 0.149 (0.00) | 0.243 (0.00) | 0.148 (0.00) | 0.241 (0.00) | 0.156 (0.00) | 0.260 (0.00) | 0.153 (0.00) | 0.250 (0.00) | 0.194 (0.00) | 0.295 (0.00) | 0.212 (0.00) | 0.327 (0.00) |
| | 336 | 0.164 (0.00) | 0.253 (0.00) | 0.171 (0.00) | 0.266 (0.00) | 0.165 (0.00) | 0.259 (0.00) | 0.165 (0.00) | 0.259 (0.00) | 0.173 (0.00) | 0.281 (0.00) | 0.169 (0.00) | 0.267 (0.00) | 0.197 (0.00) | 0.299 (0.00) | 0.215 (0.01) | 0.328 (0.01) |
| | 720 | 0.186 (0.00) | 0.275 (0.00) | 0.206 (0.01) | 0.299 (0.01) | 0.205 (0.00) | 0.295 (0.00) | 0.202 (0.00) | 0.292 (0.00) | 0.206 (0.00) | 0.308 (0.00) | 0.203 (0.000) | 0.301 (0.000) | 0.236 (0.03) | 0.329 (0.02) | 0.254 (0.01) | 0.360 (0.01) |
| ETTh1 | 96 | 0.365 (0.00) | 0.385 (0.00) | 0.404 (0.00) | 0.419 (0.00) | 0.369 (0.00) | 0.394 (0.00) | 0.375 (0.00) | 0.400 (0.00) | 0.387 (0.00) | 0.413 (0.01) | 0.379 (0.01) | 0.403 (0.01) | 0.443 (0.01) | 0.453 (0.01) | 0.496 (0.00) | 0.492 (0.01) |
| | 192 | 0.405 (0.00) | 0.411 (0.00) | 0.451 (0.00) | 0.449 (0.00) | 0.407 (0.00) | 0.415 (0.00) | 0.413 (0.00) | 0.420 (0.00) | 0.428 (0.01) | 0.439 (0.01) | 0.415 (0.01) | 0.427 (0.01) | 0.486 (0.01) | 0.482 (0.01) | 0.532 (0.04) | 0.509 (0.02) |
| | 336 | 0.437 (0.01) | 0.433 (0.01) | 0.471 (0.00) | 0.465 (0.00) | 0.392 (0.00) | 0.413 (0.01) | 0.427 (0.00) | 0.432 (0.00) | 0.505 (0.01) | 0.501 (0.01) | 0.470 (0.03) | 0.469 (0.03) | 0.482 (0.01) | 0.478 (0.01) | 0.544 (0.03) | 0.523 (0.01) |
| | 720 | 0.448 (0.01) | 0.462 (0.00) | 0.565 (0.02) | 0.538 (0.01) | 0.450 (0.00) | 0.461 (0.00) | 0.444 (0.00) | 0.463 (0.00) | 0.741 (0.09) | 0.658 (0.03) | 0.524 (0.02) | 0.527 (0.01) | 0.534 (0.03) | 0.515 (0.02) | 0.662 (0.14) | 0.592 (0.06) |
| ETTh2 | 96 | 0.276 (0.00) | 0.332 (0.00) | 0.305 (0.00) | 0.361 (0.00) | 0.261 (0.00) | 0.333 (0.00) | 0.275 (0.00) | 0.336 (0.00) | 0.370 (0.01) | 0.436 (0.01) | 0.283 (0.00) | 0.347 (0.00) | 0.363 (0.03) | 0.408 (0.02) | 0.517 (0.05) | 0.534 (0.04) |
| | 192 | 0.344 (0.00) | 0.377 (0.00) | 0.389 (0.01) | 0.411 (0.00) | 0.322 (0.00) | 0.377 (0.00) | 0.339 (0.00) | 0.379 (0.00) | 0.494 (0.03) | 0.510 (0.02) | 0.366 (0.02) | 0.403 (0.01) | 0.411 (0.02) | 0.437 (0.01) | 0.565 (0.09) | 0.559 (0.05) |
| | 336 | 0.377 (0.01) | 0.406 (0.00) | 0.420 (0.01) | 0.434 (0.01) | 0.315 (0.00) | 0.377 (0.00) | 0.328 (0.00) | 0.381 (0.00) | 0.586 (0.02) | 0.559 (0.01) | 0.428 (0.02) | 0.450 (0.01) | 0.394 (0.02) | 0.438 (0.01) | 0.757 (0.14) | 0.642 (0.07) |
| | 720 | 0.406 (0.01) | 0.435 (0.00) | 0.436 (0.01) | 0.454 (0.00) | 0.429 (0.00) | 0.453 (0.00) | 0.378 (0.00) | 0.420 (0.00) | 0.837 (0.07) | 0.688 (0.03) | 0.610 (0.06) | 0.555 (0.03) | 0.429 (0.03) | 0.458 (0.01) | 1.115 (0.17) | 0.751 (0.06) |
| ETTm1 | 96 | 0.283 (0.00) | 0.330 (0.00) | 0.305 (0.00) | 0.358 (0.00) | 0.296 (0.00) | 0.348 (0.00) | 0.290 (0.00) | 0.341 (0.00) | 0.308 (0.01) | 0.358 (0.01) | 0.301 (0.00) | 0.346 (0.00) | 0.330 (0.00) | 0.373 (0.00) | 0.497 (0.04) | 0.487 (0.02) |
| | 192 | 0.320 (0.00) | 0.354 (0.00) | 0.343 (0.00) | 0.380 (0.00) | 0.348 (0.00) | 0.378 (0.00) | 0.333 (0.00) | 0.369 (0.00) | 0.347 (0.01) | 0.386 (0.01) | 0.337 (0.00) | 0.368 (0.00) | 0.430 (0.05) | 0.423 (0.02) | 0.591 (0.03) | 0.528 (0.01) |
| | 336 | 0.357 (0.00) | 0.377 (0.00) | 0.380 (0.00) | 0.402 (0.00) | 0.376 (0.00) | 0.395 (0.00) | 0.367 (0.00) | 0.391 (0.00) | 0.398 (0.01) | 0.420 (0.01) | 0.374 (0.00) | 0.392 (0.00) | 0.397 (0.00) | 0.416 (0.00) | 0.682 (0.06) | 0.561 (0.02) |
| | 720 | 0.418 (0.01) | 0.412 (0.00) | 0.441 (0.00) | 0.438 (0.00) | 0.430 (0.00) | 0.421 (0.00) | 0.417 (0.00) | 0.422 (0.00) | 0.474 (0.03) | 0.470 (0.02) | 0.427 (0.00) | 0.423 (0.00) | 0.450 (0.01) | 0.446 (0.01) | 0.703 (0.09) | 0.579 (0.03) |
| ETTm2 | 96 | 0.162 (0.00) | 0.244 (0.00) | 0.179 (0.00) | 0.268 (0.00) | 0.170 (0.00) | 0.257 (0.00) | 0.165 (0.00) | 0.254 (0.00) | 0.182 (0.01) | 0.293 (0.01) | 0.166 (0.00) | 0.258 (0.00) | 0.190 (0.01) | 0.277 (0.00) | 0.332 (0.03) | 0.392 (0.02) |
| | 192 | 0.223 (0.00) | 0.285 (0.00) | 0.242 (0.00) | 0.312 (0.00) | 0.225 (0.00) | 0.297 (0.00) | 0.222 (0.00) | 0.293 (0.00) | 0.284 (0.03) | 0.386 (0.03) | 0.224 (0.00) | 0.301 (0.00) | 0.247 (0.01) | 0.313 (0.00) | 0.380 (0.08) | 0.414 (0.04) |
| | 336 | 0.274 (0.00) | 0.320 (0.00) | 0.292 (0.00) | 0.344 (0.00) | 0.283 (0.00) | 0.335 (0.00) | 0.277 (0.00) | 0.329 (0.00) | 0.499 (0.04) | 0.535 (0.02) | 0.280 (0.00) | 0.339 (0.01) | 0.312 (0.02) | 0.354 (0.01) | 0.432 (0.05) | 0.452 (0.03) |
| | 720 | 0.354 (0.00) | 0.373 (0.00) | 0.380 (0.01) | 0.397 (0.00) | 0.367 (0.00) | 0.386 (0.00) | 0.364 (0.00) | 0.383 (0.00) | 0.818 (0.03) | 0.695 (0.01) | 0.397 (0.01) | 0.416 (0.00) | 0.413 (0.01) | 0.411 (0.00) | 0.547 (0.09) | 0.506 (0.05) |
| Exchange | 96 | 0.088 (0.00) | 0.210 (0.00) | 0.099 (0.00) | 0.227 (0.00) | 0.169 (0.00) | 0.305 (0.00) | 0.093 (0.00) | 0.213 (0.00) | 0.113 (0.00) | 0.259 (0.01) | 0.084 (0.00) | 0.203 (0.00) | 0.167 (0.01) | 0.305 (0.01) | 0.541 (0.12) | 0.560 (0.07) |
| | 192 | 0.186 (0.00) | 0.308 (0.00) | 0.202 (0.00) | 0.326 (0.00) | 0.276 (0.00) | 0.389 (0.00) | 0.192 (0.00) | 0.312 (0.00) | 0.237 (0.03) | 0.378 (0.01) | 0.164 (0.01) | 0.293 (0.00) | 0.309 (0.02) | 0.415 (0.01) | 0.956 (0.24) | 0.770 (0.12) |
| | 336 | 0.350 (0.01) | 0.428 (0.01) | 0.397 (0.01) | 0.466 (0.01) | 0.449 (0.00) | 0.505 (0.00) | 0.350 (0.00) | 0.431 (0.00) | 0.443 (0.08) | 0.519 (0.03) | 0.355 (0.01) | 0.453 (0.00) | 0.487 (0.02) | 0.534 (0.01) | 1.290 (0.23) | 0.903 (0.08) |
| | 720 | 0.888 (0.03) | 0.689 (0.01) | 0.947 (0.01) | 0.740 (0.00) | 1.206 (0.02) | 0.821 (0.01) | 0.906 (0.00) | 0.713 (0.00) | 0.671 (0.11) | 0.653 (0.04) | 0.927 (0.05) | 0.727 (0.02) | 1.197 (0.07) | 0.842 (0.02) | 1.254 (0.03) | 0.866 (0.01) |
| Traffic | 96 | 0.358 (0.00) | 0.240 (0.01) | 0.356 (0.00) | 0.258 (0.00) | 0.398 (0.00) | 0.275 (0.00) | 0.367 (0.00) | 0.250 (0.00) | 0.488 (0.00) | 0.381 (0.00) | 0.410 (0.00) | 0.282 (0.00) | 0.605 (0.00) | 0.330 (0.00) | 0.682 (0.02) | 0.415 (0.02) |
| | 192 | 0.386 (0.01) | 0.256 (0.00) | 0.376 (0.00) | 0.268 (0.00) | 0.412 (0.00) | 0.280 (0.00) | 0.385 (0.00) | 0.259 (0.00) | 0.524 (0.01) | 0.403 (0.00) | 0.423 (0.00) | 0.287 (0.00) | 0.616 (0.00) | 0.333 (0.01) | 0.669 (0.03) | 0.411 (0.02) |
| | 336 | 0.398 (0.01) | 0.262 (0.00) | 0.389 (0.00) | 0.274 (0.00) | 0.424 (0.00) | 0.287 (0.00) | 0.399 (0.00) | 0.267 (0.00) | 0.556 (0.01) | 0.423 (0.01) | 0.436 (0.000) | 0.296 (0.000) | 0.624 (0.01) | 0.335 (0.01) | 0.674 (0.03) | 0.415 (0.01) |
| | 720 | 0.434 (0.00) | 0.284 (0.00) | 0.426 (0.00) | 0.293 (0.00) | 0.452 (0.00) | 0.305 (0.00) | 0.439 (0.01) | 0.292 (0.01) | 0.596 (0.01) | 0.444 (0.01) | 0.466 (0.000) | 0.315 (0.000) | 0.650 (0.01) | 0.346 (0.00) | 0.661 (0.02) | 0.406 (0.01) |
| Weather | 96 | 0.148 (0.00) | 0.194 (0.01) | 0.163 (0.00) | 0.212 (0.00) | 0.152 (0.00) | 0.206 (0.00) | 0.151 (0.00) | 0.199 (0.00) | 0.146 (0.00) | 0.220 (0.00) | 0.174 (0.00) | 0.235 (0.00) | 0.165 (0.00) | 0.223 (0.00) | 0.295 (0.01) | 0.370 (0.01) |
| | 192 | 0.195 (0.00) | 0.239 (0.01) | 0.207 (0.00) | 0.253 (0.00) | 0.197 (0.00) | 0.247 (0.00) | 0.196 (0.00) | 0.242 (0.00) | 0.192 (0.00) | 0.267 (0.00) | 0.216 (0.00) | 0.274 (0.00) | 0.216 (0.00) | 0.267 (0.00) | 0.382 (0.04) | 0.431 (0.03) |
| | 336 | 0.252 (0.01) | 0.282 (0.01) | 0.256 (0.00) | 0.291 (0.00) | 0.246 (0.00) | 0.285 (0.00) | 0.248 (0.00) | 0.283 (0.00) | 0.240 (0.00) | 0.304 (0.01) | 0.262 (0.00) | 0.314 (0.00) | 0.278 (0.01) | 0.309 (0.01) | 0.424 (0.04) | 0.445 (0.02) |
| | 720 | 0.323 (0.00) | 0.331 (0.01) | 0.330 (0.00) | 0.340 (0.00) | 0.327 (0.00) | 0.338 (0.00) | 0.319 (0.00) | 0.335 (0.00) | 0.311 (0.01) | 0.359 (0.01) | 0.325 (0.00) | 0.365 (0.00) | 0.338 (0.00) | 0.349 (0.00) | 0.493 (0.06) | 0.476 (0.03) |

Table 7: The full evaluation results on long-term forecasting datasets. We evaluate each model five times and take the mean of the final results. The optimal models are marked in red, and we underline the models that are not significantly worse than the optimal.

| | | DARTS-TS | | iTransformer | | ModernTCN | | PatchTST | | TSMixer | | DLinear | | TimesNet | | Autoformer | |
|---|---|---|---|---|---|---|---|---|---|---|---|---|---|---|---|---|---|
| | | MSE | MAE | MSE | MAE | MSE | MAE | MSE | MAE | MSE | MAE | MSE | MAE | MSE | MAE | MSE | MAE |
| PEMS03 | 12 | 0.066 (0.00) | 0.171 (0.00) | 0.069 (0.00) | 0.175 (0.00) | 0.112 (0.00) | 0.221 (0.00) | 0.079 (0.00) | 0.187 (0.00) | 0.075 (0.00) | 0.187 (0.00) | 0.105 (0.00) | 0.220 (0.00) | 0.085 (0.00) | 0.192 (0.00) | 0.277 (0.06) | 0.387 (0.04) |
| | 24 | 0.097 (0.00) | 0.206 (0.00) | 0.098 (0.00) | 0.209 (0.00) | 0.173 (0.00) | 0.281 (0.00) | 0.124 (0.00) | 0.235 (0.00) | 0.113 (0.00) | 0.238 (0.01) | 0.182 (0.00) | 0.296 (0.00) | 0.110 (0.00) | 0.216 (0.00) | 0.422 (0.05) | 0.466 (0.03) |
| | 48 | 0.152 (0.01) | 0.257 (0.01) | 0.448 (0.57) | 0.416 (0.28) | 0.307 (0.00) | 0.395 (0.00) | 0.223 (0.00) | 0.319 (0.00) | 0.195 (0.02) | 0.320 (0.02) | 0.318 (0.00) | 0.410 (0.00) | 0.168 (0.01) | 0.263 (0.00) | 0.806 (0.08) | 0.679 (0.04) |
| | 96 | 0.234 (0.02) | 0.331 (0.02) | 1.215 (0.62) | 0.831 (0.25) | 1.041 (0.02) | 0.779 (0.01) | 0.368 (0.00) | 0.425 (0.00) | 0.266 (0.01) | 0.380 (0.01) | 0.450 (0.00) | 0.507 (0.00) | 0.242 (0.01) | 0.321 (0.00) | 0.710 (0.15) | 0.634 (0.07) |
| PEMS04 | 12 | 0.073 (0.00) | 0.176 (0.00) | 0.081 (0.00) | 0.188 (0.00) | 0.132 (0.00) | 0.245 (0.00) | 0.101 (0.00) | 0.209 (0.00) | 0.085 (0.00) | 0.195 (0.00) | 0.115 (0.00) | 0.228 (0.00) | 0.088 (0.00) | 0.197 (0.00) | 0.562 (0.06) | 0.577 (0.03) |
| | 24 | 0.091 (0.00) | 0.198 (0.00) | 0.124 (0.00) | 0.232 (0.00) | 0.244 (0.00) | 0.338 (0.00) | 0.161 (0.00) | 0.267 (0.00) | 0.112 (0.01) | 0.228 (0.01) | 0.189 (0.00) | 0.299 (0.00) | 0.104 (0.00) | 0.216 (0.00) | 0.637 (0.10) | 0.617 (0.05) |
| | 48 | 0.120 (0.00) | 0.232 (0.00) | 0.135 (0.00) | 0.248 (0.00) | 0.452 (0.00) | 0.482 (0.00) | 0.294 (0.00) | 0.369 (0.00) | 0.159 (0.01) | 0.278 (0.01) | 0.323 (0.00) | 0.407 (0.00) | 0.138 (0.00) | 0.252 (0.01) | 1.002 (0.10) | 0.775 (0.04) |
| | 96 | 0.165 (0.00) | 0.278 (0.00) | 0.169 (0.00) | 0.280 (0.00) | 1.127 (0.00) | 0.818 (0.00) | 0.507 (0.00) | 0.505 (0.00) | 0.190 (0.01) | 0.313 (0.01) | 0.428 (0.00) | 0.484 (0.00) | 0.179 (0.00) | 0.291 (0.00) | 0.853 (0.24) | 0.708 (0.08) |
| PEMS07 | 12 | 0.060 (0.00) | 0.155 (0.00) | 0.066 (0.00) | 0.164 (0.00) | 0.085 (0.00) | 0.196 (0.00) | 0.076 (0.00) | 0.180 (0.00) | 0.070 (0.00) | 0.177 (0.00) | 0.100 (0.00) | 0.215 (0.00) | 0.083 (0.00) | 0.183 (0.00) | 0.201 (0.02) | 0.330 (0.02) |
| | 24 | 0.081 (0.00) | 0.180 (0.00) | 0.087 (0.00) | 0.190 (0.00) | 0.127 (0.00) | 0.245 (0.00) | 0.127 (0.00) | 0.234 (0.00) | 0.105 (0.01) | 0.221 (0.01) | 0.189 (0.00) | 0.302 (0.00) | 0.101 (0.00) | 0.204 (0.00) | 0.304 (0.04) | 0.402 (0.03) |
| | 48 | 0.113 (0.01) | 0.218 (0.01) | 0.892 (0.12) | 0.764 (0.08) | 0.267 (0.01) | 0.380 (0.01) | 0.238 (0.00) | 0.325 (0.00) | 0.157 (0.01) | 0.265 (0.00) | 0.375 (0.00) | 0.436 (0.00) | 0.133 (0.00) | 0.236 (0.00) | 0.422 (0.13) | 0.472 (0.08) |
| | 96 | 0.156 (0.02) | 0.262 (0.01) | 0.972 (0.19) | 0.789 (0.12) | 0.736 (0.02) | 0.673 (0.01) | 0.394 (0.00) | 0.432 (0.00) | 0.268 (0.02) | 0.342 (0.02) | 0.579 (0.00) | 0.540 (0.00) | 0.211 (0.06) | 0.308 (0.06) | 0.519 (0.10) | 0.546 (0.05) |
| PEMS08 | 12 | 0.074 (0.00) | 0.175 (0.00) | 0.089 (0.00) | 0.193 (0.00) | 0.125 (0.00) | 0.239 (0.00) | 0.091 (0.00) | 0.195 (0.00) | 0.095 (0.00) | 0.203 (0.00) | 0.112 (0.00) | 0.223 (0.00) | 0.110 (0.00) | 0.208 (0.00) | 0.467 (0.07) | 0.503 (0.05) |
| | 24 | 0.107 (0.01) | 0.213 (0.01) | 0.138 (0.00) | 0.243 (0.00) | 0.238 (0.00) | 0.336 (0.00) | 0.144 (0.00) | 0.247 (0.00) | 0.150 (0.01) | 0.257 (0.01) | 0.195 (0.00) | 0.299 (0.00) | 0.139 (0.00) | 0.234 (0.00) | 0.503 (0.07) | 0.512 (0.05) |
| | 48 | 0.175 (0.02) | 0.277 (0.02) | 0.237 (0.01) | 0.277 (0.01) | 0.528 (0.00) | 0.534 (0.00) | 0.254 (0.00) | 0.332 (0.00) | 0.256 (0.01) | 0.344 (0.01) | 0.382 (0.00) | 0.431 (0.00) | 0.194 (0.00) | 0.277 (0.00) | 0.964 (0.23) | 0.729 (0.11) |
| | 96 | 0.318 (0.04) | 0.360 (0.04) | 0.346 (0.07) | 0.363 (0.05) | 1.150 (0.00) | 0.808 (0.00) | 0.435 (0.00) | 0.441 (0.00) | 0.399 (0.02) | 0.415 (0.02) | 0.634 (0.00) | 0.550 (0.01) | 0.322 (0.01) | 0.349 (0.01) | 1.021 (0.14) | 0.763 (0.06) |

Table 8: The full evaluation results on PEMS datasets. The optimal models are marked in red, and we underline the models that are not significantly worse than the optimal.

While for the PEMS datasets (Table 10), the optimal baselines concentrates mostly on the iTransformer (Liu et al., 2024b) and TimesNet (Wu et al., 2023). However, DARTS-TS is still significantly better than the baseline models on the PEMS04 and PEMS07 datasets.

## D Comparing with zero-shot forecasting foundation models

Time series forecasting foundation models (Ansari et al., 2024; Bian et al., 2024; Das et al., 2023; Liu et al., 2024a) have become the trend for recent forecasting tasks. Instead of training multiple networks for different tasks separately, forecasting foundation models train one single model across multiple datasets and directly test it on the target sequence without further training the model on that dataset.

Here, we compare DARTS-TS with another two forecasting foundation models: TimesFM (Das et al., 2023) (with 200M parameters) and Moirai-MOE (Liu et al., 2024a) (moirai-moe-1.0-R-small with 117M parameters and 11M active parameters). Both approaches consider the input series independently and patchify each single variant time series by integrating neighboring data points into one single patch (Nie et al., 2023). Since no further training process is required for these two models, and the evaluation costs can be expensive, we only evaluate these two models with one random seed.

The result is shown in Table 11 and Table 12. We run all the foundation models on Nvidia H100 GPUs with 80GB GPU memory for at most 24 hours. Any run that fails to finish the prediction (either by memory out or out of time) will be marked with $NaN$. DARTS-TS achieves the optimal performance on the ETT datasets and is only slightly worse than TimesFM on the ECL datasets with forecasting horizons of 96 and 192. TimesFM achieves a better performance on the traffic, weather, and ECL datasets. However, these

| | | | DARTS-TS | Baseline Name | Baseline Value | statistic | p-value | | | | DARTS-TS | Baseline Name | Baseline Value | statistic | p-value |
|---|---|---|---|---|---|---|---|---|---|---|---|---|---|---|---|
| ECL | 96 | MSE | 0.129 (0.00) | PatchTST | 0.130 (0.00) | -2.114 | 0.067 | ETTm2 | 96 | MSE | 0.162 (0.00) | PatchTST | 0.165 (0.00) | -3.616 | 0.007 |
| | | MAE | 0.217 (0.00) | PatchTST | 0.223 (0.00) | -3.954 | 0.004 | | | MAE | 0.244 (0.00) | PatchTST | 0.254 (0.00) | -14.964 | 0.000 |
| | 192 | MSE | 0.147 (0.00) | PatchTST | 0.148 (0.00) | -3.985 | 0.004 | | 192 | MSE | 0.223 (0.00) | PatchTST | 0.222 (0.00) | 0.735 | 0.484 |
| | | MAE | 0.234 (0.00) | PatchTST | 0.241 (0.00) | -7.983 | 0.000 | | | MAE | 0.285 (0.00) | PatchTST | 0.293 (0.00) | -12.254 | 0.000 |
| | 336 | MSE | 0.164 (0.00) | PatchTST | 0.165 (0.00) | -1.445 | 0.186 | | 336 | MSE | 0.274 (0.00) | PatchTST | 0.277 (0.00) | -2.276 | 0.052 |
| | | MAE | 0.253 (0.00) | PatchTST | 0.259 (0.00) | -2.992 | 0.017 | | | MAE | 0.320 (0.00) | PatchTST | 0.329 (0.00) | -32.523 | 0.000 |
| | 720 | MSE | 0.186 (0.00) | PatchTST | 0.202 (0.00) | -8.918 | 0.000 | | 720 | MSE | 0.354 (0.00) | PatchTST | 0.364 (0.00) | -7.163 | 0.000 |
| | | MAE | 0.275 (0.00) | PatchTST | 0.292 (0.00) | -15.057 | 0.000 | | | MAE | 0.373 (0.00) | PatchTST | 0.383 (0.00) | -10.913 | 0.000 |
| ETTh1 | 96 | MSE | 0.365 (0.00) | ModernTCN | 0.369 (0.00) | -1.735 | 0.121 | Exchange | 96 | MSE | 0.088 (0.00) | DLinear | 0.084 (0.00) | 2.040 | 0.076 |
| | | MAE | 0.385 (0.00) | ModernTCN | 0.394 (0.00) | -5.077 | 0.001 | | | MAE | 0.210 (0.00) | DLinear | 0.203 (0.00) | 3.046 | 0.016 |
| | 192 | MSE | 0.405 (0.00) | ModernTCN | 0.407 (0.00) | -1.129 | 0.291 | | 192 | MSE | 0.186 (0.00) | DLinear | 0.164 (0.01) | 6.302 | 0.000 |
| | | MAE | 0.411 (0.00) | ModernTCN | 0.415 (0.00) | -2.168 | 0.062 | | | MAE | 0.308 (0.00) | DLinear | 0.293 (0.00) | 5.449 | 0.001 |
| | 336 | MSE | 0.437 (0.01) | ModernTCN | 0.392 (0.00) | 11.610 | 0.000 | | 336 | MSE | 0.350 (0.01) | PatchTST | 0.350 (0.00) | 0.010 | 0.992 |
| | | MAE | 0.433 (0.01) | ModernTCN | 0.413 (0.00) | 6.234 | 0.000 | | | MAE | 0.428 (0.01) | PatchTST | 0.431 (0.00) | -0.724 | 0.490 |
| | 720 | MSE | 0.448 (0.01) | PatchTST | 0.444 (0.00) | 0.920 | 0.385 | | 720 | MSE | 0.888 (0.03) | TSMixer | 0.671 (0.11) | 3.802 | 0.005 |
| | | MAE | 0.462 (0.00) | ModernTCN | 0.461 (0.00) | 0.331 | 0.749 | | | MAE | 0.689 (0.01) | TSMixer | 0.653 (0.04) | 1.634 | 0.141 |
| ETTh2 | 96 | MSE | 0.276 (0.00) | ModernTCN | 0.264 (0.00) | 25.684 | 0.000 | Traffic | 96 | MSE | 0.358 (0.00) | iTransformer | 0.356 (0.00) | 1.327 | 0.221 |
| | | MAE | 0.332 (0.00) | ModernTCN | 0.333 (0.00) | -2.560 | 0.034 | | | MAE | 0.240 (0.01) | PatchTST | 0.250 (0.00) | -3.321 | 0.011 |
| | 192 | MSE | 0.344 (0.00) | ModernTCN | 0.322 (0.00) | 18.238 | 0.000 | | 192 | MSE | 0.386 (0.01) | iTransformer | 0.376 (0.00) | 2.652 | 0.029 |
| | | MAE | 0.377 (0.00) | ModernTCN | 0.377 (0.00) | 1.436 | 0.189 | | | MAE | 0.256 (0.00) | PatchTST | 0.259 (0.00) | -1.174 | 0.274 |
| | 336 | MSE | 0.377 (0.01) | ModernTCN | 0.315 (0.00) | 22.508 | 0.000 | | 336 | MSE | 0.398 (0.01) | iTransformer | 0.389 (0.00) | 3.337 | 0.010 |
| | | MAE | 0.406 (0.00) | ModernTCN | 0.377 (0.00) | 23.442 | 0.000 | | | MAE | 0.262 (0.00) | PatchTST | 0.267 (0.00) | -2.009 | 0.079 |
| | 720 | MSE | 0.406 (0.01) | PatchTST | 0.378 (0.00) | 10.461 | 0.000 | | 720 | MSE | 0.434 (0.00) | iTransformer | 0.426 (0.00) | 1.963 | 0.107 |
| | | MAE | 0.435 (0.00) | PatchTST | 0.420 (0.00) | 7.311 | 0.000 | | | MAE | 0.284 (0.00) | PatchTST | 0.292 (0.01) | -1.488 | 0.175 |
| ETTm1 | 96 | MSE | 0.283 (0.00) | PatchTST | 0.290 (0.00) | -2.776 | 0.024 | Weather | 96 | MSE | 0.148 (0.00) | TSMixer | 0.146 (0.00) | 1.586 | 0.151 |
| | | MAE | 0.330 (0.00) | PatchTST | 0.341 (0.00) | -6.752 | 0.000 | | | MAE | 0.194 (0.00) | PatchTST | 0.199 (0.00) | -0.880 | 0.404 |
| | 192 | MSE | 0.320 (0.00) | PatchTST | 0.333 (0.00) | -4.973 | 0.001 | | 192 | MSE | 0.195 (0.00) | TSMixer | 0.192 (0.00) | 1.331 | 0.220 |
| | | MAE | 0.354 (0.00) | DLinear | 0.368 (0.00) | -9.211 | 0.000 | | | MAE | 0.239 (0.01) | PatchTST | 0.242 (0.00) | -0.482 | 0.643 |
| | 336 | MSE | 0.357 (0.00) | PatchTST | 0.367 (0.00) | -5.200 | 0.001 | | 336 | MSE | 0.252 (0.01) | TSMixer | 0.240 (0.00) | 2.741 | 0.025 |
| | | MAE | 0.377 (0.00) | PatchTST | 0.391 (0.00) | -20.458 | 0.000 | | | MAE | 0.282 (0.01) | PatchTST | 0.283 (0.00) | -0.102 | 0.921 |
| | 720 | MSE | 0.418 (0.01) | PatchTST | 0.417 (0.00) | 0.267 | 0.797 | | 720 | MSE | 0.323 (0.00) | TSMixer | 0.311 (0.01) | 2.937 | 0.019 |
| | | MAE | 0.412 (0.00) | ModernTCN | 0.421 (0.00) | -9.198 | 0.000 | | | MAE | 0.331 (0.01) | PatchTST | 0.335 (0.00) | -0.941 | 0.374 |

Table 9: Statistical Test against the non-NAS architectures on the LTSF datasets

| | | | DARTS-TS | Baseline Name | Baseline Value | statistic | p-value | | | | DARTS-TS | Baseline Name | Baseline Value | statistic | p-value |
|---|---|---|---|---|---|---|---|---|---|---|---|---|---|---|---|
| PEMS03 | 12 | MSE | 0.066 (0.00) | iTransformer | 0.069 (0.00) | -2.449 | 0.040 | PEMS07 | 12 | MSE | 0.060 (0.00) | iTransformer | 0.066 (0.00) | -6.855 | 0.000 |
| | | MAE | 0.171 (0.00) | iTransformer | 0.175 (0.00) | -2.651 | 0.029 | | | MAE | 0.155 (0.00) | iTransformer | 0.164 (0.00) | -4.497 | 0.002 |
| | 24 | MSE | 0.097 (0.00) | iTransformer | 0.098 (0.00) | -0.679 | 0.516 | | 24 | MSE | 0.081 (0.00) | iTransformer | 0.087 (0.00) | -4.173 | 0.003 |
| | | MAE | 0.206 (0.00) | iTransformer | 0.209 (0.00) | -1.589 | 0.151 | | | MAE | 0.180 (0.00) | iTransformer | 0.190 (0.00) | -4.539 | 0.002 |
| | 48 | MSE | 0.152 (0.01) | TimesNet | 0.168 (0.01) | -2.215 | 0.058 | | 48 | MSE | 0.113 (0.01) | TimesNet | 0.133 (0.00) | -3.977 | 0.004 |
| | | MAE | 0.257 (0.01) | TimesNet | 0.263 (0.00) | -1.166 | 0.277 | | | MAE | 0.218 (0.01) | TimesNet | 0.236 (0.00) | -3.299 | 0.011 |
| | 96 | MSE | 0.234 (0.02) | TimesNet | 0.242 (0.01) | -0.588 | 0.573 | | 96 | MSE | 0.156 (0.02) | TimesNet | 0.211 (0.06) | -1.799 | 0.110 |
| | | MAE | 0.331 (0.02) | TimesNet | 0.321 (0.00) | 1.170 | 0.276 | | | MAE | 0.262 (0.01) | TimesNet | 0.308 (0.06) | -1.555 | 0.158 |
| PEMS04 | 12 | MSE | 0.073 (0.00) | iTransformer | 0.081 (0.00) | -15.978 | 0.000 | PEMS08 | 12 | MSE | 0.074 (0.00) | iTransformer | 0.089 (0.00) | -11.044 | 0.000 |
| | | MAE | 0.176 (0.00) | iTransformer | 0.188 (0.00) | -16.158 | 0.000 | | | MAE | 0.175 (0.00) | iTransformer | 0.193 (0.00) | -7.653 | 0.000 |
| | 24 | MSE | 0.091 (0.00) | TimesNet | 0.104 (0.00) | -8.493 | 0.000 | | 24 | MSE | 0.107 (0.01) | iTransformer | 0.138 (0.00) | -9.285 | 0.000 |
| | | MAE | 0.198 (0.00) | TimesNet | 0.216 (0.00) | -7.884 | 0.000 | | | MAE | 0.213 (0.01) | TimesNet | 0.234 (0.00) | -4.376 | 0.002 |
| | 48 | MSE | 0.120 (0.00) | iTransformer | 0.135 (0.00) | -8.544 | 0.000 | | 48 | MSE | 0.178 (0.02) | TimesNet | 0.194 (0.00) | -1.710 | 0.126 |
| | | MAE | 0.232 (0.00) | iTransformer | 0.248 (0.00) | -7.388 | 0.000 | | | MAE | 0.277 (0.02) | iTransformer | 0.277 (0.01) | -0.016 | 0.988 |
| | 96 | MSE | 0.165 (0.00) | iTransformer | 0.169 (0.00) | -2.519 | 0.036 | | 96 | MSE | 0.318 (0.04) | TimesNet | 0.322 (0.01) | -0.160 | 0.877 |
| | | MAE | 0.278 (0.00) | iTransformer | 0.280 (0.00) | -1.439 | 0.188 | | | MAE | 0.360 (0.04) | TimesNet | 0.349 (0.01) | 0.643 | 0.538 |

Table 10: Statistical Test against the non-NAS architectures on the PEMS datasets

three datasets are also part of the training set from TimesFM, and therefore, the forecasting results on these datasets can no longer be considered as a zero-shot forecasting setup. While for the other dataset, where TimesFM is not trained, including the four PEMS datasets, DARTS-TS still outperforms TimesFM's zero-shot forecasting results.

We also provide a comparison of the latency cost during inference time between DARTS-TS, TimesFM, and Moirai-MOE. Due to the memory limitation, we only evaluate with a batch size of 1 and set the number of samples for Moirai-MoE to 10 instead of 100 random samples in its original setup. The result is shown in Figure 11. DARTS-TS provides a 67 times speed up on Traffic and a 27 times speed up on ECL compared to TimesFM. This speed-up increases to 322 times on Traffic and 123 times on ECL compared to Moirai-MoE. Showing the efficiency of DARTS-TS compared to the LLM-based approaches.

| | | DARTS-TS | | TimesFM | | Moirai-MoE | | | | DARTS-TS | | TimesFM | | Moirai-MoE | |
|---|---|---|---|---|---|---|---|---|---|---|---|---|---|---|---|
| | | MSE | MAE | MSE | MAE | MSE | MAE | | | MSE | MAE | MSE | MAE | MSE | MAE |
| ECL | 96 | 0.129 | 0.217 | 0.126 | 0.217 | 0.213 | 0.280 | ETTm2 | 96 | 0.162 | 0.244 | 0.208 | 0.273 | 0.256 | 0.315 |
| | 192 | 0.147 | 0.234 | 0.146 | 0.236 | 0.228 | 0.296 | | 192 | 0.223 | 0.285 | 0.287 | 0.323 | 0.319 | 0.353 |
| | 336 | 0.164 | 0.253 | 0.168 | 0.257 | NaN | NaN | | 336 | 0.274 | 0.320 | 0.361 | 0.369 | 0.368 | 0.382 |
| | 720 | 0.186 | 0.275 | 0.214 | 0.295 | NaN | NaN | | 720 | 0.354 | 0.373 | 0.460 | 0.434 | 0.464 | 0.432 |
| ETTh1 | 96 | 0.365 | 0.385 | 0.453 | 0.412 | 0.472 | 0.415 | Exchange | 96 | 0.088 | 0.210 | 0.109 | 0.231 | 0.083 | 0.201 |
| | 192 | 0.405 | 0.411 | 0.490 | 0.439 | 0.544 | 0.450 | | 192 | 0.186 | 0.308 | 0.228 | 0.340 | 0.184 | 0.302 |
| | 336 | 0.437 | 0.433 | 0.518 | 0.457 | 0.612 | 0.479 | | 336 | 0.350 | 0.428 | 0.412 | 0.465 | 0.350 | 0.424 |
| | 720 | 0.448 | 0.462 | 0.512 | 0.480 | 0.621 | 0.508 | | 720 | 0.888 | 0.689 | 1.035 | 0.754 | 0.897 | 0.715 |
| ETTh2 | 96 | 0.276 | 0.332 | 0.326 | 0.353 | 0.345 | 0.356 | Traffic | 96 | 0.358 | 0.240 | 0.340 | 0.225 | NaN | NaN |
| | 192 | 0.344 | 0.377 | 0.392 | 0.398 | 0.413 | 0.404 | | 192 | 0.386 | 0.256 | 0.368 | 0.241 | NaN | NaN |
| | 336 | 0.377 | 0.406 | 0.422 | 0.423 | 0.429 | 0.426 | | 336 | 0.398 | 0.262 | 0.391 | 0.254 | NaN | NaN |
| | 720 | 0.406 | 0.435 | 0.462 | 0.462 | 0.441 | 0.439 | | 720 | 0.434 | 0.284 | 0.432 | 0.279 | NaN | NaN |
| ETTm1 | 96 | 0.283 | 0.330 | 0.363 | 0.371 | 1.097 | 0.606 | Weather | 96 | 0.148 | 0.194 | 0.136 | 0.169 | 0.244 | 0.250 |
| | 192 | 0.320 | 0.354 | 0.427 | 0.411 | 1.009 | 0.600 | | 192 | 0.195 | 0.239 | 0.177 | 0.209 | 0.291 | 0.289 |
| | 336 | 0.357 | 0.377 | 0.482 | 0.447 | 0.940 | 0.592 | | 336 | 0.252 | 0.282 | 0.236 | 0.257 | 0.351 | 0.330 |
| | 720 | 0.418 | 0.412 | 0.544 | 0.487 | 0.906 | 0.594 | | 720 | 0.323 | 0.331 | 0.340 | 0.337 | 0.418 | 0.376 |

Table 11: Results against forecasting foundation models in Long Term Forecasting datasets

| | | DARTS-TS | | TimesFM | | Moirai-MoE | | | | DARTS-TS | | TimesFM | | Moirai-MoE | |
|---|---|---|---|---|---|---|---|---|---|---|---|---|---|---|---|
| | | MSE | MAE | MSE | MAE | MSE | MAE | | | MSE | MAE | MSE | MAE | MSE | MAE |
| PEMS03 | 12 | 0.066 | 0.171 | 0.158 | 0.266 | 0.124 | 0.232 | PEMS07 | 12 | 0.060 | 0.155 | 0.149 | 0.260 | 0.110 | 0.218 |
| | 24 | 0.097 | 0.206 | 0.326 | 0.391 | 0.250 | 0.330 | | 24 | 0.081 | 0.180 | 0.340 | 0.397 | 0.232 | 0.316 |
| | 48 | 0.152 | 0.257 | 0.767 | 0.632 | 0.589 | 0.526 | | 48 | 0.113 | 0.218 | 0.824 | 0.647 | 0.593 | 0.521 |
| | 96 | 0.234 | 0.331 | 1.673 | 1.008 | 1.378 | 0.867 | | 96 | 0.156 | 0.262 | 1.701 | 1.009 | 1.445 | 0.871 |
| PEMS04 | 12 | 0.073 | 0.176 | 0.176 | 0.282 | 0.134 | 0.241 | PEMS08 | 12 | 0.074 | 0.175 | 0.160 | 0.270 | 0.128 | 0.234 |
| | 24 | 0.091 | 0.198 | 0.363 | 0.416 | 0.261 | 0.340 | | 24 | 0.107 | 0.213 | 0.331 | 0.397 | 0.247 | 0.331 |
| | 48 | 0.120 | 0.232 | 0.854 | 0.670 | 0.616 | 0.543 | | 48 | 0.178 | 0.277 | 0.814 | 0.649 | 0.597 | 0.536 |
| | 96 | 0.165 | 0.278 | 1.824 | 1.055 | 1.476 | 0.903 | | 96 | 0.318 | 0.360 | 1.855 | 1.050 | 1.500 | 0.902 |

Table 12: Results against forecasting foundation models in PEMS dataset

# E  Ablation Study

## E.1  The impact of window size

In our experiments, we fixed our window size to 336 and applied this window size to predict different forecasting horizons. However, since the look-back window size is an important hyperparameter in forecasting tasks, it is interesting to see if the model should always stick to the window where it is trained. To answer this, we ask the optimizer to search for an architecture that requires the input window size $\{96, 192, 336, 720\}$. We then evaluate each of these found architectures with the window sizes mentioned above.

We run this task on the ECL-96 dataset. The result is shown in Figure 12. We see that our model will perform better with the increase in search window size in general. While the search window size also influences the final evaluations, a model searched with a window size of 720 performs the worst when the architecture suggested by this optimizer is asked to make a prediction with a window size of 96 and vice versa. However, this gap becomes smaller if the search window size is closer to the eval window size. This indicates that the window size we used to search for the network should not be too far away from the actual window size used to evaluate the model.

## E.2  Forecasting Horizon

In our experiments, we only searched with the forecasting horizon of 96 and applied them to the other forecasting horizon tasks. Here, we check how the optimal architecture searched for one target forecasting horizon can be transferred to another forecasting component. Similar to Section E.1, we ask the optimizer to search for the models with forecasting horizon of size $\{96, 192, 336, 720\}$. We then evaluate each of these optimal architectures with the other forecasting horizons mentioned above. The sliding window size is set 96 for all tasks in this scenario. The result is shown in Figure 13. The search forecasting horizons could also be transferred to the other forecasting horizons during evaluation, as different searching forecasting horizons provide similar performance under the same evaluation forecasting horizon.

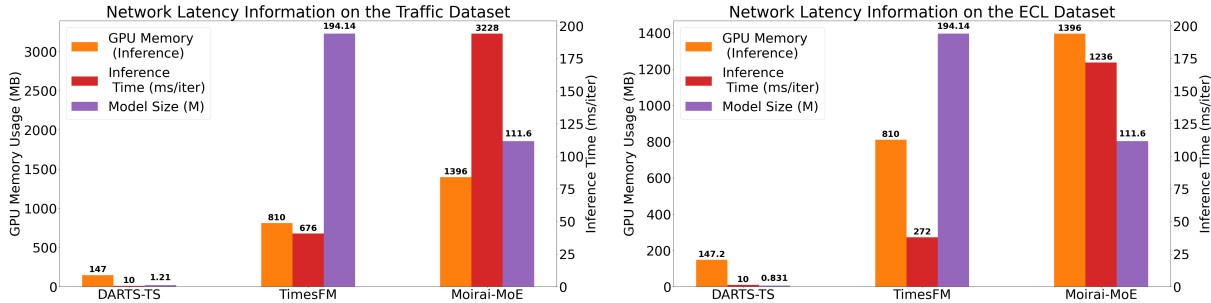

Figure 11: Latency information of different networks on the Traffic (left) and ECL (right) dataset. We set the look-back window size and the forecast horizon as 96 and the batch size as 1.

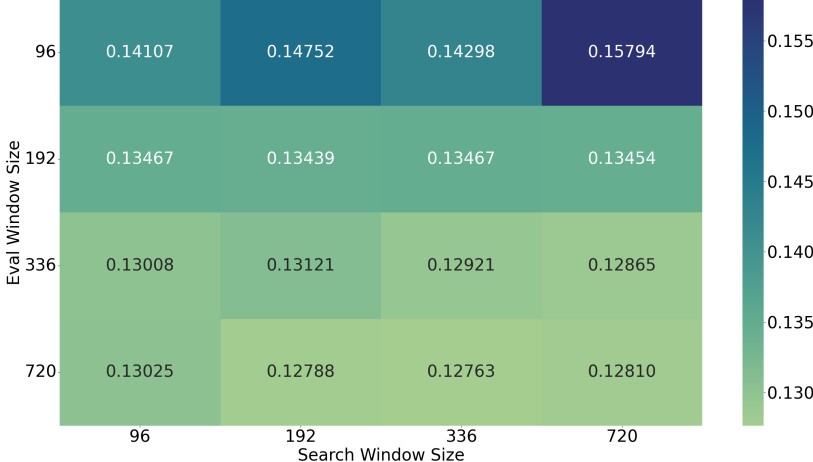

Figure 12: The impact of window size during searching and evaluations on the ECL-96 task

### E.3 Forecasting components

We constructed a hierarchical search space in Section 4. Here, we will show the efficiency of each component. We provide the following variations:

- DARTS-TS CV. This variation has the same architecture as DARTS-TS. However, the training validation dataset splits strategy during its search phase is 5-fold cross-validation instead of the holdout strategy described in Section 5

- Flat Only. This variation only contains the *Flat Net* in the search space.

- Seq Only. This variation only contains the *Seq Net* in the search space.

- Parallel. This variation is similar to our approach. However, the *Seq Net* receives only feature variables instead of the output of the *Flat Net*

- Concat Seq First. This variation first feeds the input data to the *Seq Net* and then the output of *Seq Net* is fed to the *Flat Net*

- No Weights. This variation removed the weighted sum approach described in Section 4.3

The result is shown in Table 13. Despite that DARTS-TS CV achieves nearly the same performance on some datasets, such as ECL, ETTm2, Exchange, and Weather, there is still a performance gap between DARTS-TS Holdout (i.e., DARTS-TS in the table) and DARTS-TS CV on the other datasets. The Parallel

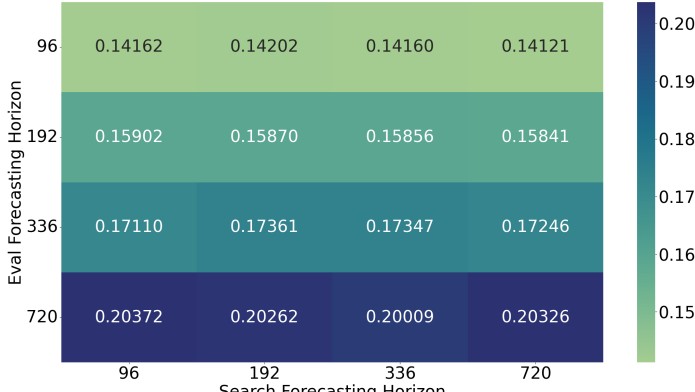

Figure 13: The impact of forecasting horizon during searching and evaluations on the ECL-96 task

| | | DARTS-TS | | DARTS-TS CV | | Parallel | | Seq Only | | Flat Only | | Concat | Seq First | | No Weights | |
|---|---|---|---|---|---|---|---|---|---|---|---|---|---|---|---|---|
| | | MSE | MAE | MSE | MAE | MSE | MAE | MSE | MAE | MSE | MAE | MSE | MAE | MSE | MAE | MSE MAE |
| ECL | 96 | 0.129 (0.00) | 0.217 (0.0) | 0.129 (0.00) | 0.220 (0.00) | 0.128 (0.00) | 0.219 (0.00) | 0.206 (0.05) | 0.305 (0.04) | 0.128 (0.00) | 0.221 (0.00) | 0.128 (0.00) | 0.221 (0.00) | 0.139 (0.01) | 0.234 (0.02) | 0.139 (0.01) 0.234 (0.02) |
| ETTh1 | 96 | 0.365 (0.00) | 0.385 (0.0) | 0.402 (0.02) | 0.420 (0.02) | 0.371 (0.01) | 0.392 (0.01) | 0.466 (0.04) | 0.463 (0.02) | 0.383 (0.00) | 0.406 (0.00) | 0.383 (0.00) | 0.406 (0.00) | 0.364 (0.00) | 0.385 (0.00) | 0.366 (0.00) 0.385 (0.00) |
| ETTh2 | 96 | 0.276 (0.00) | 0.332 (0.0) | 0.284 (0.00) | 0.339 (0.00) | 0.282 (0.00) | 0.341 (0.00) | 0.382 (0.02) | 0.421 (0.01) | 0.312 (0.00) | 0.361 (0.00) | 0.312 (0.00) | 0.361 (0.00) | 0.283 (0.00) | 0.338 (0.00) | 0.279 (0.00) 0.338 (0.01) |
| ETTm1 | 96 | 0.283 (0.00) | 0.330 (0.0) | 0.291 (0.01) | 0.335 (0.01) | 0.287 (0.00) | 0.338 (0.00) | 0.366 (0.02) | 0.402 (0.02) | 0.295 (0.00) | 0.345 (0.00) | 0.295 (0.00) | 0.345 (0.00) | 0.283 (0.00) | 0.331 (0.00) | 0.286 (0.00) 0.334 (0.01) |
| ETTm2 | 96 | 0.162 (0.00) | 0.244 (0.0) | 0.163 (0.00) | 0.245 (0.00) | 0.166 (0.00) | 0.249 (0.00) | 0.221 (0.01) | 0.301 (0.01) | 0.173 (0.00) | 0.255 (0.00) | 0.173 (0.00) | 0.255 (0.00) | 0.164 (0.00) | 0.246 (0.00) | 0.162 (0.00) 0.246 (0.00) |
| Exchange | 96 | 0.088 (0.00) | 0.210 (0.0) | 0.088 (0.00) | 0.209 (0.00) | 0.098 (0.00) | 0.220 (0.00) | 0.227 (0.03) | 0.347 (0.02) | 0.107 (0.01) | 0.234 (0.01) | 0.107 (0.01) | 0.234 (0.01) | 0.091 (0.00) | 0.213 (0.00) | 0.092 (0.00) 0.214 (0.00) |
| Traffic | 96 | 0.358 (0.00) | 0.240 (0.0) | 0.364 (0.01) | 0.246 (0.01) | 0.370 (0.02) | 0.244 (0.01) | 0.572 (0.02) | 0.312 (0.01) | 0.359 (0.00) | 0.249 (0.00) | 0.359 (0.00) | 0.249 (0.00) | 0.417 (0.01) | 0.275 (0.02) | 0.452 (0.00) 0.300 (0.01) |
| Weather | 96 | 0.148 (0.00) | 0.194 (0.0) | 0.149 (0.00) | 0.195 (0.01) | 0.150 (0.00) | 0.195 (0.01) | 0.191 (0.00) | 0.244 (0.00) | 0.151 (0.00) | 0.201 (0.00) | 0.151 (0.00) | 0.201 (0.00) | 0.149 (0.00) | 0.188 (0.00) | 0.157 (0.00) 0.209 (0.00) |

Table 13: Ablation over the components in DARTS-TS

approach is slightly worse than DARTS-TS on many datasets. However, there is a huge gap between Parallel and DARTS-TS on the Traffic dataset. While the Flat Only approach is generally worse than DARTS-TS on the ETT datasets. Overall, we show that the architectural design of DARTS-TS generally provides us with architectures that are robust across many datasets.

### E.4 Optimizazion Epochs

In Table 5, we optimize the supernet for 40 epochs and then prune the network with the pre-trained architectures. Here, we study the impact of the number of optimization epochs.

The result is shown in Figure 14. Each model is optimized for $\{0, 20, 40, 60, 80, 100\}$ epochs, respectively. Since we will also update the architecture weights and parameters during the pruning stage, even optimizing the models for zero epochs might also return a well-performing model. However, training the model for too long might overfit the validation set (Zela et al., 2020). Overall, the optimal number of epochs might differ across different datasets, while optimizing the super net for 40 or 60 epochs might result in a better average performance.

### E.5 Operations

As described in Section A, our search space contains many different operations. To verify the importance of each operation, here we remove each operation from the search space and check its performance on each of the target datasets.

The result is shown in Table 14. Despite that, removing certain operations from the search space might provide better results on certain tasks. For instance, removing Separated TCN on ECL datasets results in a model with 0.128 MSE loss instead of the 0.129 MSE loss found by the Full search space. However, removing the same operation on the ETTh1 dataset increases the loss from 0.365 to 0.381. This degeneration also appears for other operations, for instance, removing MLP increases the loss on the Traffic dataset from 0.358 to 0.379, while removing N-BEATS-Trend could further increase this value to 0.402. Hence, the search space

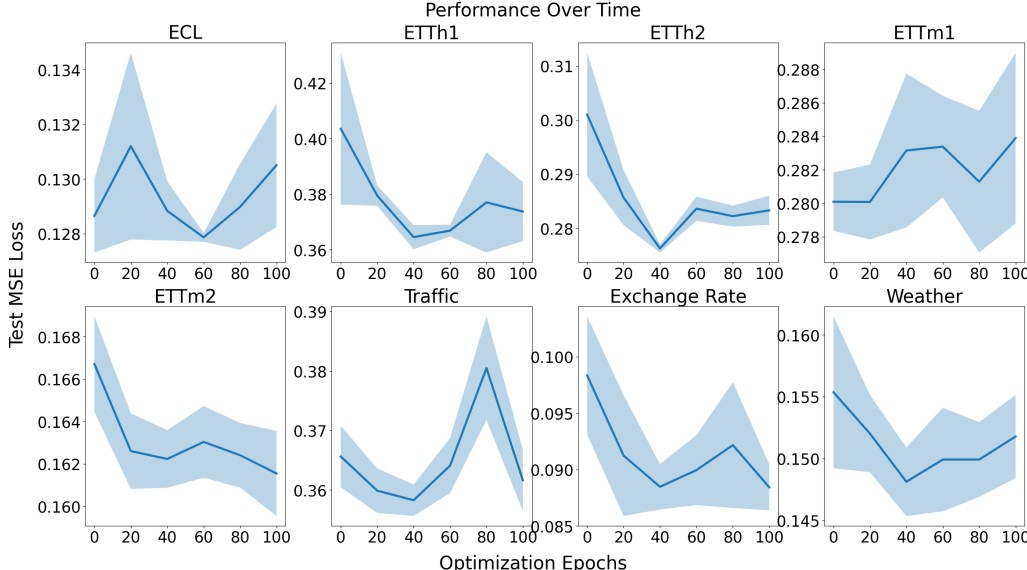

Figure 14: Performance Over Number of Supernet Optimization Epochs

| | Full Ops | | MLP | | N-BEATS G | | N-BEATS S | | N-BEATS T | | GRU | | LSTM | | MLP-Mixer | | Sep. TCN | | TCN | | Transformer | |
|---|---|---|---|---|---|---|---|---|---|---|---|---|---|---|---|---|---|---|---|---|---|---|
| | MSE | MAE | MSE | MAE | MSE | MAE | MSE | MAE | MSE | MAE | MSE | MAE | MSE | MAE | MSE | MAE | MSE | MAE | MSE | MAE | MSE | MAE |
| ECL | 0.129 | 0.217 | 0.131 | 0.223 | 0.130 | 0.222 | 0.133 | 0.227 | 0.136 | 0.233 | 0.132 | 0.224 | 0.131 | 0.221 | 0.129 | 0.220 | 0.128 | 0.216 | 0.131 | 0.220 | 0.129 | 0.219 |
| ETTh1 | 0.365 | 0.385 | 0.390 | 0.408 | 0.385 | 0.401 | 0.378 | 0.398 | 0.378 | 0.399 | 0.372 | 0.393 | 0.393 | 0.409 | 0.371 | 0.391 | 0.381 | 0.398 | 0.366 | 0.387 | 0.384 | 0.403 |
| ETTh2 | 0.276 | 0.332 | 0.289 | 0.341 | 0.286 | 0.339 | 0.286 | 0.338 | 0.291 | 0.347 | 0.292 | 0.345 | 0.292 | 0.346 | 0.286 | 0.340 | 0.286 | 0.340 | 0.287 | 0.340 | 0.287 | 0.341 |
| ETTm1 | 0.283 | 0.330 | 0.284 | 0.333 | 0.278 | 0.327 | 0.278 | 0.327 | 0.284 | 0.336 | 0.280 | 0.328 | 0.285 | 0.332 | 0.285 | 0.333 | 0.284 | 0.332 | 0.282 | 0.330 | 0.283 | 0.329 |
| ETTm2 | 0.162 | 0.244 | 0.163 | 0.245 | 0.161 | 0.245 | 0.163 | 0.246 | 0.166 | 0.250 | 0.163 | 0.248 | 0.162 | 0.245 | 0.162 | 0.245 | 0.162 | 0.245 | 0.167 | 0.250 | 0.161 | 0.244 |
| Traffic | 0.358 | 0.240 | 0.379 | 0.251 | 0.381 | 0.258 | 0.378 | 0.255 | 0.402 | 0.278 | 0.381 | 0.244 | 0.371 | 0.245 | 0.364 | 0.238 | 0.367 | 0.235 | 0.377 | 0.244 | 0.369 | 0.247 |
| Exchange Rate | 0.088 | 0.210 | 0.095 | 0.218 | 0.093 | 0.215 | 0.094 | 0.218 | 0.105 | 0.228 | 0.089 | 0.210 | 0.090 | 0.213 | 0.089 | 0.211 | 0.091 | 0.213 | 0.092 | 0.214 | 0.089 | 0.211 |
| Weather | 0.148 | 0.194 | 0.150 | 0.198 | 0.153 | 0.207 | 0.154 | 0.208 | 0.150 | 0.198 | 0.152 | 0.195 | 0.152 | 0.197 | 0.149 | 0.192 | 0.155 | 0.201 | 0.148 | 0.192 | 0.152 | 0.195 |

Table 14: Ablation over the Operations in our search space

could benefit from all the operations described in Section A depending on the datasets to which the model is applied. This further highlights the importance of exploring various architectures for different target tasks.

## E.6 RevIN

In Section 5, we applied RevIN (Kim et al., 2022) in DARTS-TS during both the searching and testing phases; here, we present an ablation study on the RevIN for DARTS-TS.

| | | ECL | ETTh1 | ETTh2 | ETTm1 | ETTm2 | Traffic | Exchange Rate | Weather |
|---|---|---|---|---|---|---|---|---|---|
| w/ RevIN | MSE | 0.129(0.001) | 0.365(0.004) | 0.276(0.001) | 0.283(0.005) | 0.162(0.001) | 0.358(0.003) | 0.088(0.002) | 0.148(0.003) |
| | MAE | 0.217(0.003) | 0.385(0.003) | 0.332(0.000) | 0.330(0.003) | 0.244(0.001) | 0.240(0.006) | 0.210(0.002) | 0.194(0.011) |
| w/o RevIN | MSE | 0.130(0.004) | 0.400(0.022) | 0.287(0.004) | 0.288(0.002) | 0.175(0.020) | 0.375(0.009) | 0.093(0.004) | 0.147(0.003) |
| | MAE | 0.225(0.009) | 0.421(0.018) | 0.341(0.002) | 0.334(0.002) | 0.267(0.030) | 0.246(0.008) | 0.219(0.003) | 0.193(0.007) |

Table 15: Ablation for the RevIN in our search space

Table 15 shows the impact of RevIN. Except for the Weather dataset, where the mean MSE loss increases from 0.147 to 0.148 by applying RevIN, the performance of all the other datasets can be improved with RevIN, especially on the ETTs and traffic datasets. Overall, RevIN enables more accurate forecasting models within our search space.

## E.7 Separated Encoder-Decoder Search Space

In Section 4.2, we design a search space that allows the encoder-decoder pairs from the same edge to have different architecture types. This design decision enlarges the search space. Here, we demonstrate that the

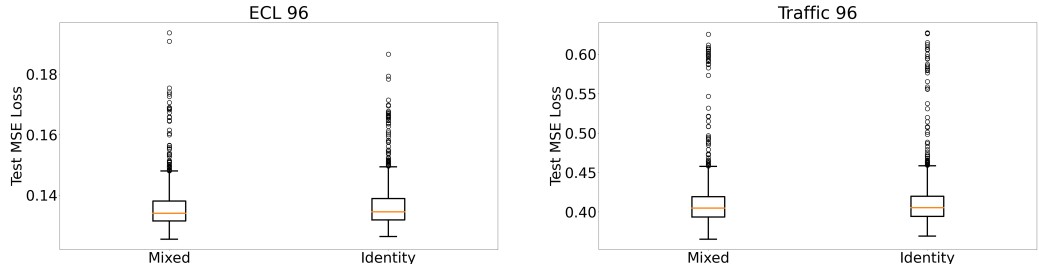

Figure 15: Performance of the architectures with mixed and identical encoder-decoder architectures

enlarged search space encompasses architectures that may outperform those with the same encoder-decoder architecture.

Hence, we check the random configurations from Section 6.2 and select all the configurations that contain *Seq* decoders. Then, for each decoder edge, instead of randomly sampling a new decoder component, we initialize the decoders with operations that are the same as those of the corresponding encoder nodes. We then reevaluate those architectures with the identical operations and compare them with the corresponding mixed architectures.

The result is shown in Figure 15. On both datasets, architectures with mixed operations achieve a lower minimum test loss (0.365 *vs* 0.369 on the Traffic dataset and 0.125 *vs* 0.126 on the ECL dataset). This shows that applying different operations to the encoders and decoders provides a potentially better model compared to the architectures that apply the same encoder and decoder operation within the search space.

## F    Optimal Architectures on other datasets

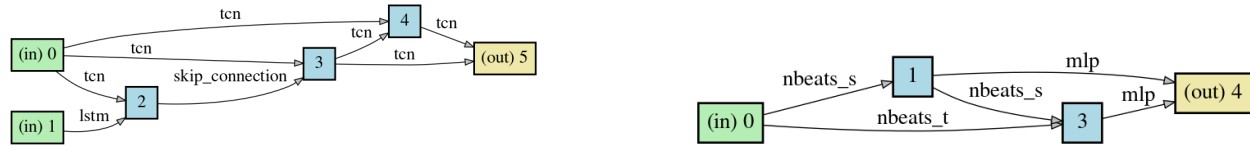

Figure 16: An optimal architecture on the ECL dataset. This is an encoder-only architecture.

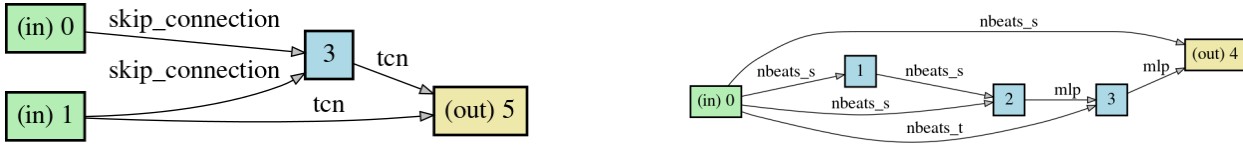

Figure 17: An optimal architecture on the Traffic dataset. This is an encoder-only architecture.

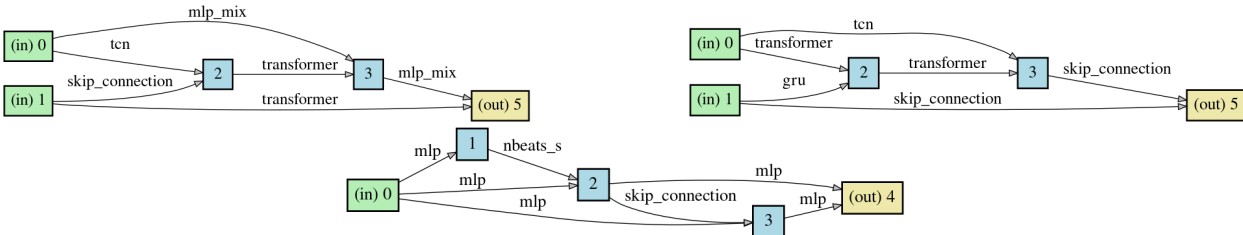

Figure 18: An optimal architecture on the ETTm1 dataset. This is an encoder-decoder architecture.

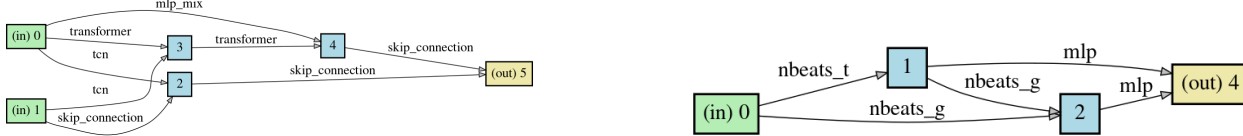

Figure 19: An optimal architecture on the ETTm2 dataset. This is an encoder-only architecture.

Figure 20: An optimal architecture on the Weather dataset. This is an encoder-only architecture.

In Section 6.4, we showed one of the optimal architectures in the electricity dataset. In this section, we will provide more searched architectures.

Figure 16 shows yet another optimal architecture on the ECL dataset. We can see that TCN and N-BEATS-Seasonal modules are still contained in the optimal modules, further confirming our conclusions in Section 6.

We also show some of the optimal architectures in Figure 17, 18, 19, and 20. We can see that no single operation dominates the optimal architecture, which shows the necessity of performing an architecture search for the optimal architecture.

