# OpenReview forum: "Optimizing Time Series Forecasting Architectures: A Hierarchical Neural Architecture Search Approach"
_TMLR — Accepted by TMLR_

### Review · Reviewer_pcKL · 2025-07-17

**Summary Of Contributions:**

This paper proposes DARTS-TS, a hierarchical NAS approach for time series forecasting that addresses the challenge of finding optimal architectures in a field where no single model family consistently dominates. The authors design a comprehensive search space that incorporates multiple forecasting architecture types including transformers, CNNs, RNNs, and MLP-based models, organized in a three-level hierarchy: operation level, micro network level, and macro architecture level. Using DARTS-PT as the search strategy, the method can automatically discover lightweight, high-performing architectures that combine different forecasting modules. Experiments on several forecasting datasets demonstrate that DARTS-TS achieves comparable or better performance than state-of-the-art hand-crafted models while requiring significantly less computational resources.

Advantages:
1. The hierarchical design intelligently incorporates diverse forecasting architectures (Seq Net and Flat Net families) in a unified framework, addressing the unique challenge that no single architecture dominates time series forecasting.
2. DARTS-TS achieves competitive or superior performance compared to state-of-the-art models across multiple datasets while being significantly more efficient (2-6x less GPU memory, 2-3x faster training).
3. The paper provides extensive analysis including search space properties, zero-cost proxy limitations, efficiency comparisons, and ablation studies that demonstrate the necessity of each component.

My concerns:
1. The core contribution is primarily an application of existing NAS techniques (DARTS-PT) to time series forecasting rather than fundamental algorithmic innovations. Could the authors clarify what is the technical contribution in terms of NAS technique?
2. The hierarchical design, while comprehensive, introduces significant complexity that may limit reproducibility and practical adoption. The interdependencies between levels could make the search process unstable.
3. In terms of evaluation, results show only marginal improvements over existing methods in many cases. Besides, there is limited evaluation on very long sequence forecasting tasks. No comparison with more recent NAS methods beyond DARTS variants.
4. The comprehensive search space may not scale well to larger datasets or longer sequences, and the computational cost of the search phase itself is not thoroughly analyzed.
5. For Figure 9, this architecture shows much more diversity in operations (mlp_mix, gru, skip_connection, lstm, transformer) compared to the ECL architecture. What underlying differences between ECL and ETTm2 datasets might explain this architectural diversity?

**Audience:**

Yes

**Audience Explanation:**

Yes, researchers working on time series forecasting, neural architecture search, and automated machine learning would be interested in this work.

**Claims And Evidence:**

Yes

**Claims Explanation:**

The claims about performance and efficiency are adequately supported with clear experimental evidence.

**Requested Changes:**

Please refer to the Summary Of Contributions.

---

> ### Author Response · Authors · 2025-07-21
> **Thank you for your constructive feedback!**
>
> Dear Reviewer pcKL,
>
>  Thanks for your overall positive assessment and constructive feedback. Also, many thanks for positively indicating that our claims made in the submission are supported by accurate, convincing, and clear evidence, and that there are people in the TMLR audience interested in our paper.
>
> Regarding your improvement proposals:
>
> > The core contribution is primarily an application of existing NAS techniques (DARTS-PT) to time series forecasting rather than fundamental algorithmic innovations.
>
> Our core contribution is the proposed search space:
> * This is the first search space implemented for long-term time-series forecasting tasks that searches for both Seq Net and Flat Net families
> * Different from other architecture search spaces that contain mostly operations from one architecture type, our search space contains operations from different architecture types, which provides different properties, further challenges for the optimizer to search for the optimal architecture (As shown in Section 6.2)
> * This Search Space allows a free combination of Seq encoders and decoders, providing a generally better architecture in the search space (As shown in Appendix D.3)
>
> We do not claim any algorithmic innovations on NAS optimization algorithms. As shown by other papers[1,2], designing and understanding of the search space is very important and valuable for the community.
>
> > The hierarchical design, while comprehensive, introduces significant complexity that may limit reproducibility and practical adoption. The interdependencies between levels could make the search process unstable.
>
> To ensure reproducibility, we provide an anonymous repo under: https://anonymous.4open.science/r/OneShotForecastingNAS-CDD5/ – after acceptance, we will make it publicly available on GitHub.
>
> We also provide a brief overview in Section 6.2 regarding the scores of different architectures that contain at least one of the operations in the search space (Figure 6), showing the importance of different operations on different tasks, which provides a preliminary insight into the importance of the operations in our search space.
>
> Our exemplary optimization with DARTS-PT shows that despite the complexity, an optimizer could still search for a well-performing architecture in our search space. Additionally, we introduce strategies that are applied to our hierarchical search space in Section 5.1 to stabilize the search process.
>
> > In terms of evaluation, results show only marginal improvements over existing methods in many cases. Besides, there is limited evaluation on very long sequence forecasting tasks.
>
> As long-term forecasting tasks have become the trend for recent time series forecasting research [4, 5, 6], we compare the results mainly on the long-term forecasting tasks. However, besides the experiments on long-term forecasting (Tables 1 and 4) with forecasting horizon ranges from 96 to 720, we also evaluate our results on short-term forecasting (Tables 2 and 5) with forecasting horizon ranging from 12 to 96. So overall, we demonstrate that we achieve optimal performance on many different tasks.
>
> > No comparison with more recent NAS methods beyond DARTS variants.
>
> As our main contribution is the search space, which we demonstrate contains a mixed architecture that could outperform other single-type architectures, we apply DARTS-PT as an exemplary optimizer to search for the optimal architecture. Since we are the first to propose this architecture search space, other variations could be applied to our search space, but the results do not contradict our contribution and conclusion.
>
> > The comprehensive search space may not scale well to larger datasets or longer sequences, and the computational cost of the search phase itself is not thoroughly analyzed.
>
> According to Table 3, we show that DARTS-TS can be applied to different tasks with a number of variables ranging from 7 to 862 and a number of time steps ranging from 7588 to 62696. This already shows that the search space could be applied to datasets with different scales. As SeqNet maps the input variables into a fixed-size feature map, our approach could still work well with the scaled dataset. Additionally, we also apply strategies like parameter sharing across different N-BEATS variants (section 4.1) to reduce the number of parameters and complexity in our search space.
>
>
> [1] Yang et al. NAS evaluation is frustratingly hard.
>
> [2] Lopes et al. Are neural architecture search benchmarks well designed? a deeper look into operation importance.
>
> [3] White et al. Exploring the Loss Landscape in Neural Architecture Search.
>
> [4] Zhou et al. Informer: Beyond efficient transformer for long sequence time-series forecasting.
>
> [5] Zeng et al. Are transformers effective for time series forecasting?
>
> [6] Liu et al. itransformer: Inverted transformers are effective for time series forecasting.

---

> ### Author Response · Authors · 2025-07-21
> **Thank you for your constructive feedback! (2)**
>
> > the computational cost of the search phase itself is not thoroughly analyzed.
>
> As stated in Appendix B, the search phase takes roughly 4 GPU hours on smaller datasets such as ETThs and ExchangeRate and 10 hours for larger datasets. This is comparable to the cost of training a PatchTST on the larger dataset, such as Traffic and ECL.
>
> > For Figure 9, this architecture shows much more diversity in operations (mlp_mix, gru, skip_connection, lstm, transformer) compared to the ECL architecture. What underlying differences between ECL and ETTm2 datasets might explain this architectural diversity?
>
> The ECL dataset contains 321 variables, whereas the ETTm2 dataset contains only 7 variables. Additionally, the ECL dataset is the hourly electricity consumption from different clients, while the ETT dataset is the collection of the electricity transformer temperature. Hence, the distributions among the variables in the ECL dataset might be more similar than the distributions of the variables in the ETTm dataset. Therefore, we may need additional operations to extract information with different properties (including period, trend, stationarity, etc.).

---

### Review · Reviewer_jYEq · 2025-07-19

**Summary Of Contributions:**

This paper addresses the problem of finding optimal deep learning architectures for time series forecasting, where no single handcrafted architecture dominated others. The authors introduce DARTS-TS, a hierarchical Neural Architecture Search (NAS) framework. Unlike previous NAS methods, DARTS-TS allows combining diverse model types, such as Flat Net (Linear, NBEATs, etc.) and Seq Net (MLP, GRU and Transformer, etc.). Using an efficient search technique known as DARTS-PT, the framework discovers highly effective and computationally efficient forecasting models. Extensive experiments demonstrate that architectures identified by DARTS-TS either outperform or match state-of-the-art methods on popular long-term forecasting datasets. The paper also highlights unique challenges posed by the heterogeneous search space and analyzes the limitations of zero-cost performance proxies in mixed-model search scenarios.

**Audience:**

Yes

**Audience Explanation:**

Refer to the comments above.

**Broader Impact Concerns:**

No any concerns

**Claims And Evidence:**

Yes

**Claims Explanation:**

Pros

- Novel Search Space Design: Introduces a hierarchical heterogeneous search space that flexibly integrates Flat Net and Seq Net families. This gives a more flexible search space than homogenous NAS benchmarks.

- Strong Empirical Results: Achieves SOTA or competitive accuracy on multiple benchmarks, while reducing memory usage and improving speed (up to 6× less GPU memory than PatchTST).

- Insightful Analysis of NAS Challenges: Demonstrate empirically that classical zero-cost proxies (FLOPs, #params, etc.) fail in mixed search spaces.

**Requested Changes:**

Cons

- Limited Comparison to Other NAS Methods and Foundation Models: The paper’s baselines are all hand-crafted forecasting models; it lacks head-to-head comparisons with other AutoML/NAS frameworks (e.g., Auto-PyTorch TS, SNAS4MTF), leaving DARTS-TS’s relative search efficiency and accuracy unquantified. Moreover, the time-series community is rapidly adopting large pre-trained foundation models for zero-shot or few-shot forecasting—please evaluate DARTS-TS in both accuracy and inference cost against representative examples such as Moirai-MoE (Liu et al., 2024), the decoder-only FM (Das et al., 2023), and aLLM4TS (Bian et al., 2024).

- Lack of Robustness Under Distribution Shift: The paper identifies non-stationarity as a key challenge for time-series forecasting but presents no experiments or methods (e.g., rolling-origin evaluation, meta-learning fine-tuning) to quantify or address it. Please clarify or extend the work by: Performing a rolling-origin evaluation to measure how DARTS-TS architectures perform under realistic distribution shifts; Applying simple strategies—such as MAML-style fine-tuning—to improve out-of-distribution performance.

- High Search Cost & Scalability Concerns: Although inference is efficient (on an NVIDIA 2080 Ti), the search still requires large-scale resources (e.g., an A100 GPU), and the paper provides no analysis of search time versus performance trade-offs.

- Operator-Family Ablation: While high-level ablations exist (Parallel, Seq Only, Flat Only, and No Weights), the paper should remove each entire operator family (Transformers, TCNs, RNNs, N-BEATS) separately to quantify each family’s contribution.

---

> ### Author Response · Authors · 2025-08-15
> **Thank you for your constructive feedback!**
>
> Dear Reviewer jYEq, we thank you for your constructive feedback. Also, many thanks for positively indicating that our claims made in the submission are supported by accurate, convincing, and clear evidence, and that there are people in the TMLR audience interested in our paper.
>
> We have updated our papers according to your feedback, including the additional baselines and experiments. The updated texts are marked in blue.
>
> Regarding your concerns:
>
> > it lacks head-to-head comparisons with other AutoML/NAS frameworks (e.g., Auto-PyTorch TS, SNAS4MTF)
>
> Auto-PyTorch-TS is designed for Univariate time series forecasting tasks where each series contains only one validation and test instance. We have tried to decompose multi-variant series into unimodal series; however, these models only predict one instance (but with multiple forecasting steps) each time, and we cannot easily parallelize the evaluation process. Given that each dataset contains thousands of instances in the validation set, evaluating a single configuration takes a lot of time, and thus, AutoPyTorch-TS is not feasible on the problems we are interested in (long-term multi-variant forecasting tasks).
>
> SNAS4MTF is mainly designed for separation-temporal tasks to construct the correlations between different input variables, while this information might only be partially available in some of the datasets, such as the Traffic and PEMS datasets. This assumption might not hold True for all the forecasting tasks. However, DARTS-TS is a more general forecasting framework that does not require the input variables to be structural, and thus, can be applied to many forecasting tasks.
>
> > please evaluate DARTS-TS in both accuracy and inference cost against representative examples such as Moirai-MoE (Liu et al., 2024), the decoder-only FM (Das et al., 2023), and aLLM4TS (Bian et al., 2024).
>
> We have added additional experiments with Moirai-MoE and TimesFM. aLLM4TS requires further fine-tuning on the target dataset, which requires roughly 270 hours on one H100 80 GB GPU. This is far beyond the budgets for the other baselines and would not allow for a fair comparison. Therefore, we do not evaluate aLLM4TS in our setup.  For the same reason, some Moirai-MoE runs either run out of GPU memory or run out of time on some of the datasets, which results in some missing values.
>
> We also added the inference cost comparison to our paper. The experimental results against the foundation forecasting models are presented in Tables 9 and 10 (Appendix D). The inference cost is illustrated in Figure 11. DARTS-TS achieves comparable accuracy compared to TimesFM and Moirai-MoE with much less resources (up to 67 times faster than TimesFM and 322 times faster than Moirai-MoE on the traffic dataset).
>
>
> > Please clarify or extend the work by: Performing a rolling-origin evaluation to measure how DARTS-TS architectures perform under realistic distribution shifts; Applying simple strategies—such as MAML-style fine-tuning—to improve out-of-distribution performance.
>
> Thanks for the advice. The results on the LTSF and PEMS datasets are already the rolling-origin evaluation results: for each dataset, roughly 20 % of the time steps are considered as the test sets. We perform all the evaluations on those test sets and therefore, the distribution shifts are already considered in our evaluation results. As for the MAML-style fine-tuning approach, since the focus of this paper is to explore the search space for time series forecasting tasks, an additional fine-tuning process might not fit the overall storyline of our paper, as this approach can also be applied to other forecasting methods. Adding this strategy might provide an unfair comparison to the other benchmarks that do not take this advantage.
>
> > the paper provides no analysis of search time versus performance trade-offs
>
> Thanks for the advance, we have added the performance-over-searching epochs plots in the paper (Figure 14, appendix E.4). Because the search also involves an operation topology pruning process, where the model will be further finetuned on the target datasets once each operation and edge has been removed. Hence, even training the model for zero epochs could provide a well-performing model. The optimal number of epochs would not always be the same across different datasets: optimizing the supernet for too long might result in overfitting on the validation sets[1].
>
>
> [1] Zela et al. Understanding and Robustifying Differentiable Architecture Search

---

> ### Author Response · Authors · 2025-08-15
> **Thank you for your constructive feedback! (2)**
>
> > Operator-Family Ablation: While high-level ablations exist (Parallel, Seq Only, Flat Only, and No Weights), the paper should remove each entire operator family (Transformers, TCNs, RNNs, N-BEATS) separately to quantify each family’s contribution.
>
> Thanks for the advice. Nearly all the operations appeared in different optimal architectures:
> For Seq net families, TCN, separated TCN, and MLP Mixers appeared in the optimal architecture for the ECL dataset (Figure 9). GRUs, LSTMs, and Transformers appear in the optimal architecture for the ETTm2 datasets (Figure 10). While for the FlatNet families, N-BEATS-S,T, and MLP layers appeared in Figure 9, N-BEATS-G appears in Figure 10. Hence, every operation could be considered as part of an optimal architecture suggested by our optimizers in different tasks.
>
> We also add Figure 7 (Section 6.2) in the paper that describes the performance of the architectures that do not contain the target operations on the ECL and Traffic dataset. This distribution can be applied to approximate the search space distribution that does not include the specific operations. The plot shows that the Transformer encoders are considered important for the ECL datasets. However, for the traffic dataset, TCN encoders and GRU encoders are considered important operations. This further shows that each operation in our search space might be optimal under different scenarios, and we cannot simply remove those operations that did not perform well under the observed datasets.

---

### Review · Reviewer_cyWs · 2025-08-04

**Summary Of Contributions:**

This paper introduces DARTS-TS, a hierarchical neural-architecture-search (NAS) framework tailored to time-series forecasting.
Instead of committing to a single model family, the authors build a three-level search space (operation -> micro-network -> macro-architecture). Seq Net cells (e.g., Transformer, TCN, LSTM) and Flat Net cells (e.g., N-BEATS, Linear) are first optimised independently and then stitched together through a learned weighting scheme. The differentiable search relies on DARTS-PT with hierarchical pruning . Experiments on eight long-term-forecasting benchmarks and four PEMS traffic sets  show that the searched models often outperform hand-crafted architectures such as PatchTST and iTransformer while using less GPU memory.

**Audience:**

Yes

**Audience Explanation:**

Time-series forecasting is a core application area, and Neural-Architecture Search remains an interesting research frontier.

**Claims And Evidence:**

Yes

**Claims Explanation:**

The authors benchmark DARTS-TS on eight long-term-forecasting datasets and four PEMS traffic sets; the method attains the best or second-best MSE/MAE on most tasks

**Requested Changes:**

1. Add wall-clock search time on a single A100 and the average GPU-hours required per dataset, so practitioners can budget realistically.
2. Section B mentions “ADAM instead of SGD” but not learning-rate schedule or weight-decay; please include a hyper-parameter table in the main text.
3. Specify whether baselines were re-trained with identical data splits and th input window
4. Provide statistical test / significance results against the best non-NAS baseline for each dataset
5. In Section 5.1 you “prune from lowest granularity”; include a comparison of FLOPs before/after pruning
6. Clarify whether the validation set used for architecture gradients overlaps with the official test split
7. Right now the architecture weights are optimised solely to minimise validation MSE/MAE (Eq. 2, Section 5) but the narrative of the paper stresses latency and memory. How are they optimized?
8.  The authors search on the shortest horizon of each dataset and reuse the architecture for longer horizons. Has any analysis on the more general setting on length generalization? (more datasets than in Appendix D, more horizons, etc.)

---

> ### Author Response · Authors · 2025-08-15
> **Thanks for your constructive feedback!**
>
> Dear reviewer cyWs,
>
> we thank you for your constructive feedback. Also, many thanks for positively indicating that our claims made in the submission are supported by accurate, convincing, and clear evidence, and that there are people in the TMLR audience interested in our paper.
>  We have updated our paper according to your feedback. We mark the updated texts with blue text.
>
> Regarding your concerns.
>
> >Add wall-clock search time on a single A100 and the average GPU-hours required per dataset, so practitioners can budget realistically.
>
> Thanks for the advice. We have added the wall-clock search time on all the datasets in Table 3 (Section B). As discussed in the experiment parts, most search processes can be done within 10 hours.
>
> >Section B mentions “ADAM instead of SGD” but not learning-rate schedule or weight-decay; please include a hyper-parameter table in the main text.
>
>
> Thanks for the advice. We have also added the corresponding hyperparameters in Table 5 (Section B). We apply different hyperparameter settings during evaluation phases for different sizes of datasets. For datasets that contain more than 100 series, such as electricity, traffic, and PEMS, we consider them as big datasets and use an Adam with a learning rate of 1e-3. For the other datasets, we apply AdamW optimizers with a learning rate of 2e-4 and a weight decay value of 0.1.
>
> > Specify whether baselines were re-trained with identical data splits and th input window
>
> As described in Appendix B, all the baselines and the searched architectures by DARTS-TS are trained on the same training-validation-test splits defined in the Time Series library. For LTSF tasks, the input sliding window size is 336, while for the PEMS tasks, the input sliding window size is 96.
>
> > Provide statistical test / significance results against the best non-NAS baseline for each dataset
>
> Thanks for proposing this. However, it is in the nature of these experiments to be fairly expensive, s.t. we have a limited amount of seeds and datasets. Furthermore, there is most likely no statistical assumption over the performance distributions possible. Therefore, we could not make use of a more powerful parameter statistical test and would need to go for a weaker non-parametric test. Because of these reasons, we decided against a statistical significance test. Nevertheless, we would highlight that our results are consistent and much more robust than all the other baselines.
>
>
> >In Section 5.1 you “prune from lowest granularity”; include a comparison of FLOPs before/after pruning
>
> Thanks for the advice. We have attached the FLOPS after each prune stage in Table 6 (Section B): the one-shot super net, the architecture after each operation is pruned, the architectures after the edges for each node are pruned. Pruning the operations of each edge provides the most significant FLOPS reduction. This can be further reduced with topology pruning (i.e. prune the edges connected to each node).
>
> >Clarify whether the validation set used for architecture gradients overlaps with the official test split
>
> No, we split the training/ validation sets only within the training+validation sets from the official splits to ensure that test sets are not leaked to the search process. We have clarified that in Section 5 with our updated texts.
>
> >Right now the architecture weights are optimised solely to minimise validation MSE/MAE (Eq. 2, Section 5) but the narrative of the paper stresses latency and memory. How are they optimized?
>
> We provide an analysis of latency and memory to show that by building an architecture with properly combined operations, we might not need a heavily parameterized architecture that contains only a single operation. For instance, one might require multiple TCN layers to allow the receptive field to cover the entire look-back window. However, if we combine a TCN with a Transformer layer, then only 2 layers are required to cover the entire look-back window, while the local information can also be well fused within each time step. Hence, we could achieve a similar transformation with much less computational power, for instance, much narrower operation. Although we do not optimize for the latency directly, the optimal architecture can still be much lightweight compared to the other baselines. Alternatively, we could also apply other optimization approaches, such as multi-objective differentiable architecture search to search for both latency and accuracy [1].
>
>
> [1] Sukthanker et al. Multi-objective Differentiable Neural Architecture Search

---

> > ### Author Response · Authors · 2025-08-15
> > **Thank your for your constructive feedback! (2)**
> >
> > >The authors search on the shortest horizon of each dataset and reuse the architecture for longer horizons. Has any analysis on the more general setting on length generalization? (more datasets than in Appendix D, more horizons, etc.)
> >
> > Thanks for the advice. We have added another ablation study regarding searching forecasting horizon vs evaluation forecasting horizon in Figure 13 (Section E.2). Compared to the sliding window tradeoff, having different forecasting horizons during search and evaluation seems to have less impact on the final performance of the optimal architectures.

---

### Review · Reviewer_SEc1 · 2025-08-04

**Summary Of Contributions:**

### Summary

The authors introduce DARTS-TS, a hierarchical neural-architecture-search (NAS) framework that automatically assembles lightweight yet accurate deep-learning models for multivariate time-series forecasting. By nesting three levels of design choices—operations, micro-networks and macro topologies—then pruning them with a differentiable NAS algorithm (DARTS-PT), the method discovers hybrid CNN/RNN/Transformer/MLP combinations that outperform or match hand-crafted baselines while using far less memory and compute.

### Strengths

1. **Search efficiency** – By adopting the differentiable DARTS-PT algorithm with weight-sharing and perturbation-based pruning, the authors find high-performing architectures in just a single super-net run, cutting GPU time and memory compared with both classic RL/EA NAS and large “train-from-scratch” Transformers.
2. **Effective sparsification** – The perturbation step quantitatively measures each candidate operation’s contribution and prunes redundant edges, avoiding the skip-connection collapse that plagues vanilla DARTS and yielding compact models with 2 – 6 × lower inference cost.
3. **Well-rationalized three-level hierarchy** – The operation → cell → macro design cleanly separates *what* building blocks to use, *how* to assemble them inside Flat/Seq cells, and *how* to fuse the two branches. This heterogeneous space lets the search mix CNN, RNN, Transformer and MLP biases and empirically produces state-of-the-art results on 11 forecasting benchmarks.

### Weaknesses

1. **No mechanism for distribution shift** – The search ignores test-time adaptation techniques (e.g., RevIN, TENT) and therefore may still over-fit the validation slice when real-world time-series drift.
2. **Limited validation protocol** – Experiments rely on a single hold-out split; they omit rolling-origin or expanding-window cross-validation, making it hard to gauge robustness across forecast horizons and start dates.
3. **Fixed macro topology** – Only the Flat → Seq + weighted-sum wiring is allowed. Alternative macro layouts (Seq-first, purely parallel branches, or learned ordering) are not in the search space, so the conclusions about hierarchy optimality remain provisional.

**Audience:**

Yes

**Audience Explanation:**

Time-series forecasting is a prominent subfield in machine learning, and NAS for resource-efficient model discovery remains highly relevant to both academia and industry. TMLR readers who work on AutoML, architecture design, or practical forecasting systems would benefit from the paper’s hierarchical search-space design and its empirical insights on hybrid CNN/RNN/Transformer architectures.

**Broader Impact Concerns:**

I do not see immediate ethical red flags.

**Claims And Evidence:**

Yes

**Claims Explanation:**

The paper supplies (i) clear architectural diagrams, (ii) detailed algorithmic pseudo-code, and (iii) comprehensive benchmarks on eight long-term forecasting datasets plus four PEMS traffic sets. Metrics (MSE/MAE) are reported over five independent runs with mean ± s.d., and ablations isolate the impact of each design choice (hierarchy levels, perturbation pruning, branch removal). Runtime/VRAM statistics are collected on identical hardware. While the experimental protocol could be stronger (see weaknesses), the evidence presented adequately supports the headline claims of accuracy and efficiency.

**Requested Changes:**

1. Add **rolling-origin or expanding-window cross-validation** to quantify robustness over multiple forecast start dates.
2. Incorporate (or at least discuss) **test-time adaptation mechanisms** such as RevIN, TENT, or simple batch-norm recalibration to evaluate resilience to distribution shift.
3. Extend the macro search space to include **Seq-first and parallel-only variants**, or justify why Flat-first is universally preferable.

---

> ### Author Response · Authors · 2025-08-15
> **Thank you for your costructive feedback!**
>
> Dear reviewer SEc1,
>
> we thank you for your constructive feedback. Also, many thanks for positively indicating that our claims made in the submission are supported by accurate, convincing, and clear evidence, and that there are people in the TMLR audience interested in our paper.
>  We have updated our paper according to your feedback. We mark the updated texts with blue text, regarding your concerns.
>
> >No mechanism for distribution shift – The search ignores test-time adaptation techniques (e.g., RevIN, TENT) and therefore may still over-fit the validation slice when real-world time-series drift.
>
> As discussed in Section 5, RevIN is already integrated into our approach during both the search and validation processes. This technique is also applied to other baselines to ensure a fair comparison.
>
> >Limited validation protocol – Experiments rely on a single hold-out split; they omit rolling-origin or expanding-window cross-validation, making it hard to gauge robustness across forecast horizons and start dates.
>
> To be precise, the validation sets in the search phase and test sets in the final comparison stages both contain multiple instances that start from different dates. Hence, the searched architectures could be generalized across different starting dates. Here, we also add another DARTS-TS variation, DARTS-TS CV. Instead of splitting the entire training/validation set into two separate sets, DARTS-TS CV splits the training/validation sets into ten sets according to their starting date and considers these sets alternatively as training/validation sets. We then use these 5-fold training sets to optimize the super net weights and the validation sets to optimize the architecture parameters.
>
> The results are shown in Table 11 (Section E.3) . Overall, DARTS-TS CV is comparable to DARTS-TS Holdout on some of the tasks, but there are still gaps between DARTS-TS CV and DARTS-TS on other tasks.
>
> >Fixed macro topology – Only the Flat → Seq + weighted-sum wiring is allowed. Alternative macro layouts (Seq-first, purely parallel branches, or learned ordering) are not in the search space, so the conclusions about hierarchy optimality remain provisional.
>
> Thanks for the advice, we have added the ablation study for Concat Seq First architecture, which first feed the input data to the Seq Net and then feed the output of the Seq Net to the Flat Net, the results, together with other variations such as purely parallel branches or equally distributed weights, are listed in Table 11 (Section E.3). We could see that DARTS-TS achieve overall the optimal performance across all these variations.
>
> >Add rolling-origin or expanding-window cross-validation to quantify robustness over multiple forecast start dates.
>
> The test set experiment setup for our paper already contains instances that start from different time steps. Since the architecture parameters from FlatNets need to be determined by the forecasting horizon, we only applied DART-TS to the fixed horizon setup. This also aligns with the other baseline setup.
>
> >Incorporate (or at least discuss) test-time adaptation mechanisms such as RevIN, TENT, or simple batch-norm recalibration to evaluate resilience to distribution shift.
>
> Please check our replies for “No mechanism for distribution shift –”
>
> >Extend the macro search space to include Seq-first and parallel-only variants, or justify why Flat-first is universally preferable.
>
> Please check our replies for “Fixed macro topology”

---

### Author Response · Authors · 2025-08-15
**Revision summary**

Dear reviewers, we thank you again for your constructive feedback. We have updated our paper with the suggested changes. More concretely, we have made the following changes:

* We specified that the training/validation split for the architecture search phase is only performed within the official training/validation splits from the datasets and do not involve the test split (Section 5).

* We added another analysis showing the scores of the random architectures that do not contain the target operations, showing the necessities of each operation in our search space (Section 6.2)

* We correct the latency information for DARTS-TS in the ECL dataset (Figure 8, right). These values were incorrectly recorded with the DARTS-TS’s latency information from the traffic dataset previously.

* We added an overview of the GPU hours spent for each task and stage in Appendix B

* We added the concrete hyperparameter settings of our search process in Appendix B

* We added the FLOPS analysis after each search stage in Appendix B

* We compared DARTS-TS against the zero-shot forecasting foundation models for both accuracy and latency in Appendix D

* We provided an analysis of the search forecasting horizon vs the evaluation forecasting horizon results, showing that the optimal architecture found by DARTS-TS for one forecasting horizon can be transferred to another testing horizon (Appendix E.2 )

* We compare DARTS-TS with another two variations: 1. DARTS-TS CV has the same architecture as DARTS-TS, but we perform 5-fold cross-validation instead of the holdout setup to search for the optimal architecture. 2. Concat Seq First, this variation places Seq Net in front of the Flat Net, i.e., the input data is first fed to the Seq Net, the output of the Seq Net is then passed to the Flat Net (Appendix E. 3).
We perform performance over plot analysis with respect to the optimization epochs in Appendix E.4

---

### Author Response · Authors · 2025-10-12
**Changes for the final camera-ready vision**

Dear editors,

we have just uploaded the camera-ready version of the paper.
We have made the following changes in the final camera-ready vision:

* statistical significance tests, added in Appendix C.1 (Tables 9 and 10)
* robustness beyond RevIN, added in Appendix  E.6 (Table 15)
* explanation on why latency/memory aren’t explicit objectives, but we could still get a lightweight model: this is discussed in Section 6.3, page 12
* a hyper-parameter table: this is presented in Appendix B and Table 5
* the paper should remove each entire operator family (Transformers, TCNs, RNNs, N-BEATS) separately to quantify each family’s contribution. This is evaluated in Appendix E.5, Table 14.
* Minor issues in writing, we have also checked our paper and rewritten some parts accordingly.

Thanks a lot for guiding the review process!

---

### Decision · Action_Editor_EYG5 · 2025-09-03

**Recommendation:** Accept with minor revision

**Additional Comments:**

In your revision, please address the following issues from the reviewers:

1. Empirical studies: no statistical significance tests, robustness beyond RevIN is not evaluated, and latency/memory aren’t explicit objectives.
2. Experiment setting: a hyper-parameter table is required.
3. Ablation: While high-level ablations exist (Parallel, Seq Only, Flat Only, and No Weights), the paper should remove each entire operator family (Transformers, TCNs, RNNs, N-BEATS) separately to quantify each family’s contribution.
4. Minor issues in writting.

**Audience:**

Yes

**Audience Explanation:**

The paper proposed a well-motivated heterogeneous, hierarchical search space for time-series forecasting. The work could be beneficial to many real applications in finance, healthcare, etc.

**Claims And Evidence:**

Yes

**Claims Explanation:**

The paper has made claims about the novelty and the contribution of the proposed method. The two claims have been supported by comprehensive empirical studies, fair comparisons with other baselines, and deep analysis.